# Systemic biological mechanisms underpin poor post-discharge growth among severely wasted children with HIV

In sub-Saharan Africa, children with severe malnutrition (SM) and HIV have substantially worse outcomes than children with SM alone, facing higher mortality risk and impaired nutritional recovery post-hospitalisation. Biological mechanisms underpinning this risk remain incompletely understood. This case-control study nested within the CHAIN cohort in Kenya, Uganda, Malawi, and Burkina Faso examined effect of HIV on six months post-discharge growth among children with SM and those at risk of malnutrition, assessed proteomic signatures associated with HIV in these children, and investigated how these systemic processes impact post-discharge growth in children with SM. Using SomaScan™ assay, 7335 human plasma proteins were quantified. Linear mixed models identified HIV-associated biological processes and their associations with post-discharge growth. Using structural equation modelling, we examined directed paths explaining how HIV influences post-discharge growth. Here, we show that at baseline, HIV is associated with lower anthropometry. Additionally, HIV is associated with protein profiles indicating increased complement activation and decreased insulin-like growth factor signalling and bone mineralisation. HIV indirectly affects post-discharge growth by influencing baseline anthropometry and modulating proteins involved in bone mineralisation and humoral immune responses. These findings suggest specific biological pathways linking HIV to poor growth, offering insights for targeted interventions in this vulnerable population.

Childhood malnutrition, including wasting, underweight, and stunting remains a significant global public health concern, particularly in sub-Saharan Africa, where 27% of the 45 million children with wasting and 43% of the 148 million children with stunting live[1]. Despite progress in prevention of mother-to-child HIV transmission, the region continues to bear a high burden of paediatric HIV, accounting to over 80% of the 130,000 new HIV cases in children under 14 years in 2022[2]. Malnutrition and HIV infection often coexist in children, and malnutrition is a key clinical feature in children with HIV. A recent systematic review and meta-analysis estimated the prevalence of wasting, underweight, and stunting among children living with HIV in East Africa to be 25%, 42%, and 50% respectively[3]. HIV also increases the risk of growth failure

during the first two years of life, even in well-nourished infants after birth[4]. It may also affect post-discharge growth recovery among children at risk of malnutrition following resolution of acute illness[5].

Malnutrition in its most-life threatening form, severe malnutrition (SM), is characterised by severe muscle wasting or presence of nutritional oedema and affects a considerable proportion of children under 5 years old in sub-Saharan Africa[1]. Children presenting with SM and concomitant infection, decreased appetite or severe oedema are classified as having complicated SM, necessitating their admission to hospital for inpatient care. Reported mortality among children with complicated SM in sub-Saharan Africa remains unacceptably high, up to 10–20% in hospital and 10–15% in the year post-discharge[6–8]. A

✉ e-mail: EMudibo@kemri-wellcome.org; JNjunge@kemri-wellcome.org; Bryan.Gonzales@UGent.be

substantial fraction of children hospitalised with SM in sub-Saharan Africa also have HIV infection (HIV-SM)[9]. HIV has a major impact on the epidemiology, pathogenesis, clinical presentation, and outcomes of SM in children in many parts of Africa. Hospitalised children with HIV and SM face elevated mortality, increased morbidity, slower nutritional recovery, and greater risk of malnutrition relapse than children with SM alone[10–13]. Survivors may suffer long-term impairments in growth and physical function[14,15]. The molecular mechanisms underlying poor nutritional recovery in children with HIV-SM remain poorly understood.

HIV may exacerbate features associated with severe malnutrition such as perturbations of multiple physiological, immune, and hormonal pathways, co-infections, enteropathy, and inflammation[15–18]. We recently identified metabolic stress markers among children with HIV-SM compared to their SM counterparts[15]. Specifically, children with HIV-SM exhibited distinct metabolic perturbations, including enriched pathways related to inflammation and lipid metabolism, reduced plasma levels of zinc-alpha-2-glycoprotein, butyrylcholinesterase and increased levels of complement C2, which are often reported in the context of obesity, metabolic syndrome, and other non-communicable diseases[15]. Understanding the relation between changes in metabolic, immune, hormonal pathways and growth in children with HIV-SM is crucial to inform development of new interventions that restore physiology and induce optimal growth in this vulnerable population[19]. Thus, we aimed to firstly determine the effect of HIV on six months post-discharge growth among children with severe malnutrition and those at risk of malnutrition following acute illness in sub-Saharan Africa. Secondly, we sought to identify HIV-associated systemic pathways among these children. Lastly, we examined how HIV impacts growth through these identified biological pathways in children with severe malnutrition.

## Results

### Baseline characteristics of the study participants

Figure 1A outlines the selection of children included in this study. A total of 834 children alive at discharge were included in this HIV case-control analysis. Participant characteristics at discharge and clinical illness at admission are detailed in Table 1. Of the 834 study children alive at discharge, 112 (13.4%) had HIV. While age and sex were comparable, the majority of children with HIV were from Blantyre, Kampala, and Migori sites (Table 1). At discharge, children with HIV were more wasted, underweight, and stunted. Although common clinical presentations including diarrhoea and pneumonia were comparable, a higher proportion of children with HIV had pulmonary tuberculosis (TB) at hospital admission. One in four children with HIV were on antiretroviral treatment (ART) at admission (Table 1). To examine biological pathways through which HIV influenced 90 days post-discharge anthropometry, 38 children with HIV were compared with 179 children without HIV as illustrated in Fig. 1A.

### HIV is associated with lower anthropometry at hospital discharge

We first examined whether anthropometry was comparable among study children by HIV status at discharge. We found that at discharge children with HIV had mean mid-upper arm circumference (MUAC) of 11.13 cm (±1.58) compared to 12.15 cm (±1.66) for children without HIV. Additionally, children with HIV had mean weight-for-age z score (WAZ), weight-for-height z score (WHZ) and height-for-age z score (HAZ) of −3.46 (±1.66), −2.52 (±2.01) and −3.01 (±1.87) respectively compared to −2.47 (±1.76), −1.77 (±1.62) and −2.17 (±1.85) for children without HIV (Table 1 and Fig. 1B). We then examined whether there were differences in post-discharge anthropometric trajectories by HIV status. We anticipated that since children with HIV had larger weight deficits at discharge, they would consequently have larger weight gains post-discharge. Overall, the study children had mean monthly gains of

0.45 cm (95% CI: 0.42 to 0.47) in MUAC and z scores of 0.22 (95% CI: 0.20 to 0.24) and 0.31 (95% CI: 0.28 to 0.35) in WAZ and WHZ, respectively (Fig. 1C). Notably, weight related anthropometric gains were greatest between discharge to 45 days post-discharge compared to later periods (Fig. 1B). Consistent with our hypothesis, children with HIV had mean monthly gains of 0.18 cm (95% CI: 0.10 to 0.25) in MUAC, WAZ of 0.14 (95% CI: 0.08 to 0.21) and WHZ of 0.15 (95% CI: 0.06 to 0.24) more than those without HIV (Fig. 1C). Further, the study children had mean monthly losses of 0.06 (95% CI: −0.08 to −0.04) HAZ during the follow up period (Fig. 1B, C). However, children with HIV experienced modest linear growth, gaining a monthly mean HAZ of 0.06 (95% CI: 0.01 to 0.11) more than their counterparts post-discharge (Fig. 1C). The individual growth trajectories for every child stratified by HIV status are shown in Supplementary Fig. 1.

### Specific systemic processes are enriched among children with HIV at hospital discharge

After examining the effect of HIV on growth through 6 months post-hospitalisation, we subsequently performed analyses to understand how HIV imparts its effect on early post-discharge growth in children. Firstly, we explored systemic proteome expression profiles associated with HIV. Here, we compared 79 children with HIV to 610 without HIV. Supplementary Table 1 highlights the characteristics of these children. Weighted correlation network analysis of the 7335 plasma proteins resulted in 40 protein modules (PM), including one with unassigned proteins (herein labelled as PM40). PM sizes ranged between 11 (PM1) and 1203 (PM39) proteins (Fig. 2A). Our analysis indicated that of the 39 assigned modules, 27 were significantly associated with HIV status (Fig. 2A). We also observed that there were 6 superclusters (SC) of highly correlated HIV-associated modules (R ≥ 0.5; Fig. 2B, Table 2). Supplementary Fig. 2B shows the hierarchical clustering dendrogram of module eigenproteins and further revealed how modules related with each other. Eight protein modules (PMs 37, 31, 6, 16, 39, 2, 8 and 5) and supercluster 6 (SC6) were significantly increased while 19 modules (PMs 32, 30, 10, 25, 33, 38, 34, 17, 3, 7, 1, 26, 14, 12, 18, 27, 19, 23 and 22) and 3 superclusters (SC1, SC2 and SC3) were decreased among children with HIV compared to those without HIV (Fig. 2A). Superclusters 4 and 5 comprised both negatively and positively HIV-associated modules (Fig. 2). Enrichment analysis of proteins within these HIV-associated modules showed that HIV was positively associated with complement activation, translation initiation, peptide transport and humoral immune response. Additionally, HIV was associated with reduced expression of proteins involved in neuronal communication, blood coagulation, cellular morphogenesis, bone metabolism and growth, innate immune responses, cytokine production and messenger RNA metabolic processes as shown in Table 2. However, several individual modules including PMs 2, 3, 5, 6, 10, 14, 16 and 25 were not enriched for any biological processes. Supplementary Table 2 shows functional annotation for all modules associated with HIV status.

To gain deeper insights into these biological processes, we identified highly influential proteins within the modules, termed as hub proteins which offered valuable insights beyond enrichment analysis and determined their association with HIV status. Our analysis showed that 16 out of the 27 modules associated with HIV, demonstrated a significant correlation between the absolute effect size and connectivity within a module (Fig. 3 panels B, C, F, G, I, K, M-P, S, V, W, Y, Z and AA). Except for PM39 (Fig. 3AA), all the other 15 modules demonstrated positive correlations and exhibited significant differences in expression levels of hub protein between children with and without HIV (Fig. 3). Some of the hub proteins positively associated with HIV include trafficking protein particle complex subunit 3 (TRAPPC3) which is involved in intracellular vesicular transport and part of a complex required for HIV infection[20], sodium-coupled monocarboxylate transporter 1 (SC5A8); a sodium ion-coupled solute transporter of short-chain fatty acids, monocarboxylates

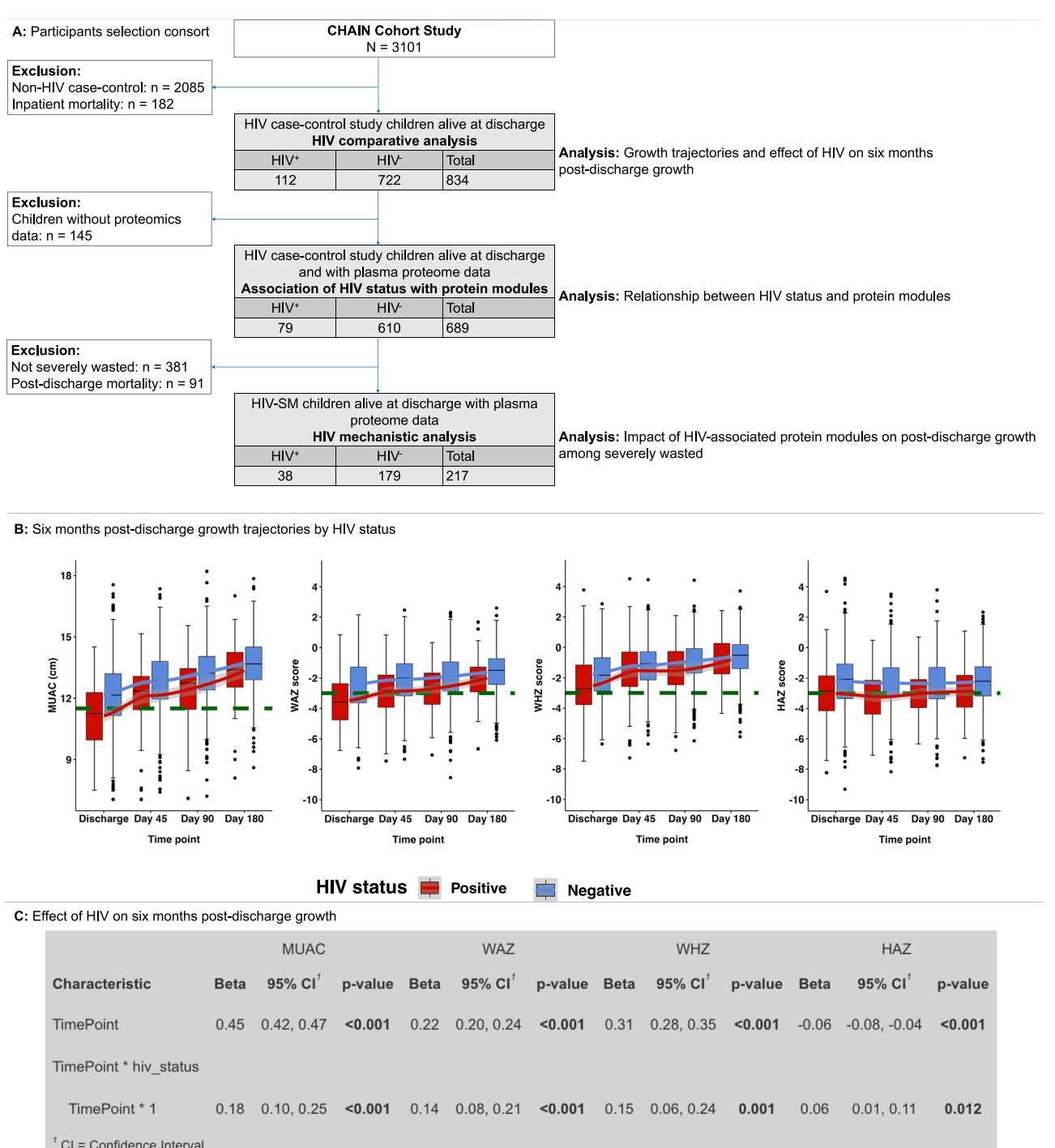

**A:** Participants selection consort

**B:** Six months post-discharge growth trajectories by HIV status

**HIV status** — Positive — Negative

**C:** Effect of HIV on six months post-discharge growth

| Characteristic | MUAC | | | WAZ | | | WHZ | | | HAZ | | |
|---|---|---|---|---|---|---|---|---|---|---|---|---|
| | Beta | 95% CI[1] | p-value | Beta | 95% CI[1] | p-value | Beta | 95% CI[1] | p-value | Beta | 95% CI[1] | p-value |
| TimePoint | 0.45 | 0.42, 0.47 | **<0.001** | 0.22 | 0.20, 0.24 | **<0.001** | 0.31 | 0.28, 0.35 | **<0.001** | -0.06 | -0.08, -0.04 | **<0.001** |
| TimePoint * hiv_status | | | | | | | | | | | | |
| TimePoint * 1 | 0.18 | 0.10, 0.25 | **<0.001** | 0.14 | 0.08, 0.21 | **<0.001** | 0.15 | 0.06, 0.24 | **0.001** | 0.06 | 0.01, 0.11 | **0.012** |

[1] CI = Confidence Interval

drugs and ketone bodies and whose expression is affected by inflammation and is increased in T-cells during HIV infections[21], and complement factor H-related protein 5 (CFHR5); a component of the complement system whose expression is increased in adults living with HIV[22] (Fig. 3, Supplementary Table 3). Serotransferrin that binds and transports iron, kininogen that plays a role in coagulation, insulin-like growth factor binding protein 3 (IGFBP-3) that binds and regulates the actions of insulin-like growth factor 1 (IGF-I), enteropeptidase involved in catalytic activation of trypsinogen to trypsin and superoxide dismutase (SOD), an antioxidant enzyme were some of the hub proteins negatively associated with HIV. In addition to the single topmost hub protein described above for each module, we provide a larger list of top hub proteins from the modules significantly associated with HIV in Supplementary Table 4.

Further, we examined the associations of proteins in these modules to identify core proteins driving network connectivity in addition to the candidate hub proteins identified above. We observed that complement components C9 and CFHR5 controlled the connectivity of PM8 which was enriched for complement activation. PM37 and PM31 which were enriched for humoral immune responses were centred around tumour necrosis factor-α (TNFα), interleukin 6 (IL-6), hypoxia-inducible factor 1-α (HIF1A) and TNF ligand superfamily member 11 (TNFSF11). PM39 linked to translation initiation was connected by charged multivesicular body proteins (CHMP3, CHMP2A, CHMP2B, CHMP6), Supplementary Fig. 3. IGF-I, IGFBP-3, lGF-acid labile subunit (IGFALS), leptin and growth hormone receptor (GHR) involved in growth factor signalling controlled the connectivity of PM7 that was enriched for insulin-like growth factor receptor signalling. PM26 that

**Fig. 1 | The HIV-SM case-control participants selection consort and post-discharge growth trajectories. A** Study participants selection consort of the HIV-SM case-control study. For the effect of HIV on post-discharge growth analysis, the study included 834 children from the CHAIN cohort of whom 112 had HIV infection. To unravel systemic proteome processes associated with HIV status, the study analysed 689 children (those with plasma proteomics data measured at hospital discharge) of whom 79 had HIV infection. Children who lacked proteomics data were excluded from this analysis. For the growth mechanistic analysis to understand impact of HIV-associated biological processes on growth, children who died post-discharge, lacked proteome data or were not severely wasted were excluded, retaining 217 children of whom 38 had HIV. **B** Depicts box plots of 180 days post-discharge mid-upper arm circumference (MUAC), weight-for-age z score (WAZ), weight-for-height z score (WHZ) and height-for-age z score (HAZ) trajectories, stratified by HIV status. Data are presented as median values with interquartile ranges (IQR). The plots include fitted LOESS (locally estimated scatterplot smoothing) function (red and blue lines) to show the post-discharge growth rates. The red and blue colours represent the HIV positive ($n = 112$) and HIV negative ($n = 722$) groups, respectively. Green horizontal dashed lines show the threshold for severe malnutrition based either on MUAC (11.5 cm) or Z score of (-3) for WAZ, WHZ, and HAZ. Box plots indicate; median (middle line); 25th (first quartile, Q1) and 75th (third quartile, Q3) percentile (box limits); error bars (whiskers) represent 1.5*Q1 and Q3 while single points outside the error bars represent outliers. **C** Shows mean post-discharge anthropometric gains overall and results from an interaction analysis between HIV status and time. Beta coefficient estimates and p-values were obtained using a fixed-effects panel model (Eq. 1), where children without HIV served as the reference group. Statistically significant results were identified based on $p < 0.05$. For each predictor, significance was assessed with t-statistics, while the F-test evaluated overall model significance, as implemented in the plm function from the plm R package. The exact $p$ values for the relationships between MUAC, WAZ, WHZ, and HAZ with TimePoint are as follows: MUAC and TimePoint, $p = 5.31e\text{-}186$; WAZ and TimePoint, $p = 6.24e\text{-}69$; WHZ and TimePoint, $p = 7.00e\text{-}73$; and HAZ and TimePoint, $p = 1.74e\text{-}10$. For the interaction effects in the model, the exact p-values are as follows: TimePoint*HIV status and MUAC, $p = 3.35e\text{-}06$; TimePoint*HIV status and WAZ, $p = 2.17e\text{-}05$. Abbreviations: Discharge, hospital discharge time point; Day 45, 45 days post-discharge; Day 90, 90 days post-discharge; Day 180, 180 days post-discharge; CI, confidence interval.

was enriched for bone mineralisation was centred around collagen alpha-1(I) chain (COL1A1), collagen alpha-2(XI) chain (COL11A2) and thrombospondin 4 (THBS4); proteins involved in development of bone, skeletal muscle and extracellular matrix. Some of these core proteins including CFHR5, TNFα and TNFSF11 were elevated while IGF-I, IGFBP-3, IGFALS, leptin, GHR, COL11A2, COL1A1 and THBS4 were decreased in children with HIV compared to those without HIV as shown in Supplementary Figs. 3 and 4. Collectively, these findings indicate a complex modulation of diverse biological processes in response to HIV infection in children. Additionally, the findings implicate that specific proteins are influential in driving the network connectivity and functionality among the modules associated with HIV.

### HIV-associated systemic pathways are linked with post-discharge growth among severely wasted children

Following the identification of the HIV-associated systemic processes, we next examined their associations with 90-day post-discharge growth among children with severe malnutrition. We included all children with severe malnutrition at discharge (MUAC < 11.5 cm, $n = 217$) of whom 38 had HIV (Fig. 1A). The characteristics of the study children included in the analysis are outlined in Table 3. In summary, common clinical illness at admission, sex and age at discharge were comparable between children with and without HIV.

We observed that of the 27 protein modules associated with HIV (Fig. 2A), PM12 (enriched for cell morphogenesis) and PM33 (enriched for ribosomal biogenesis) were positively associated with 90-day post-discharge MUAC and WHZ respectively (Fig. 4A, C). PM26 and PMs 31 and 37 which were enriched for bone mineralisation and humoral immune response respectively were all negatively associated with 90-day post-discharge MUAC, WAZ and WHZ (Fig. 4A–C). Additionally, PM7 which was enriched for IGF-I signalling was negatively associated with 90-day post-discharge WAZ and WHZ (Fig. 4B, C). PM3 which was enriched for proteins involved in polyglutamylation and PM12 were positively while PM18 (linked to regulation of activin receptor signalling pathway) was negatively associated with 90-day post-discharge HAZ. Overall, these findings indicate that HIV is associated with biological processes related to post-discharge growth. Table 2 and Supplementary Table 2 highlights significantly over-represented biological processes within modules associated with 90-day post-discharge growth.

Our analysis further showed that neugrin (NGRN), a hub protein for PM37, that is vital in mitochondrial ribosome biogenesis was associated with 90-day post-discharge MUAC (Fig. 4H). Additionally, the insulin gene enhancer protein (ISL1, PM37), a transcriptional activator that regulates the expression of insulin, glucagon, somatostatin

and pancreatic polypeptides as well as IGF-I (PM7) were associated with both 90-day post-discharge WAZ and WHZ (Fig. 4I, L, M, and P). Thrombospondin-3 (TSP3, PM26), involved in long bone and skeletal growth and stathmin-4 (STMN4, PM33) that plays a role in microtubule depolymerisation and neuron projection development were associated with 90-day post-discharge WHZ (Fig. 4N, Q). Finally, we observed that tubulin polyglutamylase complex subunit 2 (TPGS2, PM3) that is involved in protein polyglutamylation was the only hub protein associated with 90-day post-discharge HAZ (Fig. 4R). All these candidate hub proteins exhibited significant differences in their expression levels between children with and without HIV (Fig. 4). Broadly, these results indicate that specific proteins play a key role in driving the connectivity and functionality of HIV-related biological processes linked to post-discharge growth. Candidate hub proteins among HIV-related modules associated with 90-day post-discharge growth are summarised in Supplementary Table 5 while the networks of every module and the core proteins driving the connectivity of the respective network are shown in Supplementary Figs. 3 and 4.

### HIV appears to modify specific systemic processes among severely wasted children to influence early post-discharge growth

Upon identifying HIV-related systemic processes associated with early post-discharge growth among children with severe malnutrition, we investigated directed paths linking HIV to post-discharge growth in these children. We hypothesised that HIV indirectly influences early post-discharge growth via modulation of specific systemic processes (Supplementary Fig. 5). Using structural equation modelling (SEM), we investigated potential directed paths linking HIV to 90-day post-discharge growth. Overall, we found that HIV influenced 90-day post-discharge growth by modulating specific systemic biological processes (Fig. 5). However, there was no direct association between HIV and post-discharge growth across all the SEM frameworks (Fig. 5). We also observed that HIV was directly associated with baseline anthropometry only within the MUAC framework (Fig. 5A) which was consistent with our earlier findings that linked HIV with lower anthropometry at hospital discharge (Fig. 1B). As expected, baseline anthropometry was positively associated with 90-day post-discharge growth in all the models. The framework analysis for MUAC showed that HIV was associated with PMs 26, 31, and 37 but not PM12 as had been observed in the regression analysis (Fig. 5A). Here, processes related to PM26 and PM31 mediated the relationship between HIV and MUAC at 90 days post hospitalisation. Notably, we infer that HIV negatively modulated biological processes linked to bone mineralisation and collagen fibril organisation (PM26) which were in turn negatively associated with 90-day post-discharge MUAC. We also found

**Table 1 | Characteristics of the study participants at discharge**

| Characteristics at discharge | | HIV⁻ (*n* = 722) | HIV⁺ (*n* = 112) | Total (*N* = 834) |
|---|---|---|---|---|
| Demographic | | | | |
| Sex – Males N (%) | | 410 (57%) | 63 (56%) | 473 (57%) |
| Age, months – Median (IQR) | | 12.7 (8.0–17.5) | 12.1 (6.2–17.8) | 12.6 (7.9–17.5) |
| Site – *N* (%) | | | | |
| Banfora | | 128 (18%) | 4 (3.6%) | 132 (16%) |
| Blantyre | | 83 (11%) | 39 (35%) | 122 (15%) |
| Kampala | | 210 (29%) | 27 (24%) | 237 (28%) |
| Kilifi | | 88 (12%) | 13 (12%) | 101 (12%) |
| Migori | | 131 (18%) | 22 (20%) | 153 (18%) |
| Nairobi | | 82 (11%) | 7 (6.3%) | 89 (11%) |
| Nutritional status at discharge – N (%) | | | | |
| No wasting | | 276 (38%) | 22 (20%) | 298 (36%) |
| Moderate wasting | | 197 (27%) | 27 (24%) | 224 (27%) |
| Severe wasting | | 249 (34%) | 63 (56%) | 312 (37%) |
| Anthropometry at discharge | | | | |
| MUAC (cm) | Median (IQR) | 12.15 (11.15–13.20) | 11.25 (9.95–12.28) | 12.05 (10.95–13.10) |
| | Mean (SD) | 12.15 (±1.66) | 11.13 (±1.58) | 12.01 (±1.68) |
| WAZ score | Median (IQR) | −2.42 (−3.62 to −1.27) | −3.58 (−4.76 to −2.37) | −2.59 (−3.82 to −1.35) |
| | Mean (SD) | −2.47 (±1.76) | −3.46 (±1.66) | −2.60 (±1.78) |
| WHZ score | Median (IQR) | −1.82 (−2.88 to −0.68) | −2.72 (−3.76 to −1.16) | −1.90 (−3.00 to −0.77) |
| | Mean (SD) | −1.77 (±1.62) | −2.52 (±2.01) | −1.87 (±1.70) |
| HAZ score | Median (IQR) | −2.08 (−3.34 to −1.08) | −2.88 (−4.16 to −1.85) | −2.18 (−3.46 to −1.15) |
| | Mean (SD) | −2.17 (±1.85) | −3.01 (±1.87) | −2.28 (±1.87) |
| Oedema *N* (%) | | 20 (2.9%) | 3 (2.8%) | 23 (2.9%) |
| Clinical illness at admission – *N* (%) | | | | |
| Diarrhoea | | 289 (40%) | 51 (46%) | 340 (41%) |
| Pneumonia | | 358 (50%) | 52 (46%) | 410 (49%) |
| Malaria Positive (RDT) | | 140 (20%) | 6 (5.6%) | 146 (18%) |
| Measles | | 17 (2.4%) | 1 (0.9%) | 18 (2.2%) |
| Sepsis | | 39 (5.4%) | 10 (9.3%) | 49 (5.9%) |
| Pulmonary TB | | 15 (2.1%) | 12 (11%) | 27 (3.3%) |
| On Co-trimoxazole prophylaxis and Antiretroviral treatment at admission – *N* (%) | | | | |
| Co-trimoxazole prophylaxis | | 9 (1.2%) | 31 (28%) | 40 (4.8%) |
| Antiretroviral treatment (ART) | | – | 29 (26%) | – |

Data are median (IQR), mean (SD) or count, *n* (%).
*RDT* rapid diagnostic test, *TB* tuberculosis, *MUAC* mid-upper arm circumference, *WAZ* weight-for-age, *WHZ* weight-for-height, *HAZ* height-for-age, *IQR* interquartile range, *SD* Standard deviation.

that HIV positively modulated markers of humoral immune response (PM31) which was negatively associated with MUAC at discharge. On the other hand, PM37 which is a part of a supercluster linked to humoral immune response and positively associated with HIV, did not exhibit any associations with baseline or 90-day post-discharge MUAC (Fig. 5A).

Similar path analysis indicated that while PMs 7, 26, 31, and 37 were associated with HIV in the frameworks for WAZ and WHZ, only PM31 appeared to mediate the relationship between HIV and 90-day post-discharge WAZ and WHZ (Fig. 5B, C). Specifically, HIV positively modulated PM31 related to humoral immune response that in turn negatively influenced 90-day post-discharge WAZ and WHZ. Our findings also show that HIV modulated bone mineralisation (PM26) and insulin-like growth factor receptor signalling pathways (PM7) which subsequently influenced baseline but not post-discharge WAZ (Fig. 5B). Additionally, while PM37 showed positive associations with HIV, it did not exhibit any association with baseline or 90 days post-discharge WAZ and WHZ (Fig. 5B, C). Finally, we observed that HIV was negatively associated with protein polyglutamylation (PM3) which then had a positive association with 90-day post-discharge HAZ (Fig. 5D). We also observed that while PM12, linked to cellular

morphogenesis and PM18 linked to postsynaptic membrane assembly and activin receptor regulation were not modulated by HIV, they significantly influenced baseline and 90-day post-discharge HAZ respectively (Fig. 5D).

## Discussion

This study investigated the effect of HIV on six months post-discharge growth among children with severe malnutrition and those at risk of malnutrition, and assessed how HIV modulates specific systemic processes that influence early post-discharge growth among children with severe malnutrition in sub-Saharan Africa. Our analysis primarily focused on MUAC, WAZ and WHZ as these are the criteria mainly used for diagnosing malnutrition in children under 5 years[23]. The analysis revealed that children with HIV were more wasted, underweight and stunted compared to those without HIV at hospital discharge. We observed that children with HIV had insufficient catch-up growth despite experiencing greater post-discharge gains in weight and MUAC compared to those without HIV. Our results show that HIV is linked to baseline MUAC and indirectly to post-discharge growth but not directly associated with post-discharge growth. At hospital discharge, children with HIV had increased biological processes including

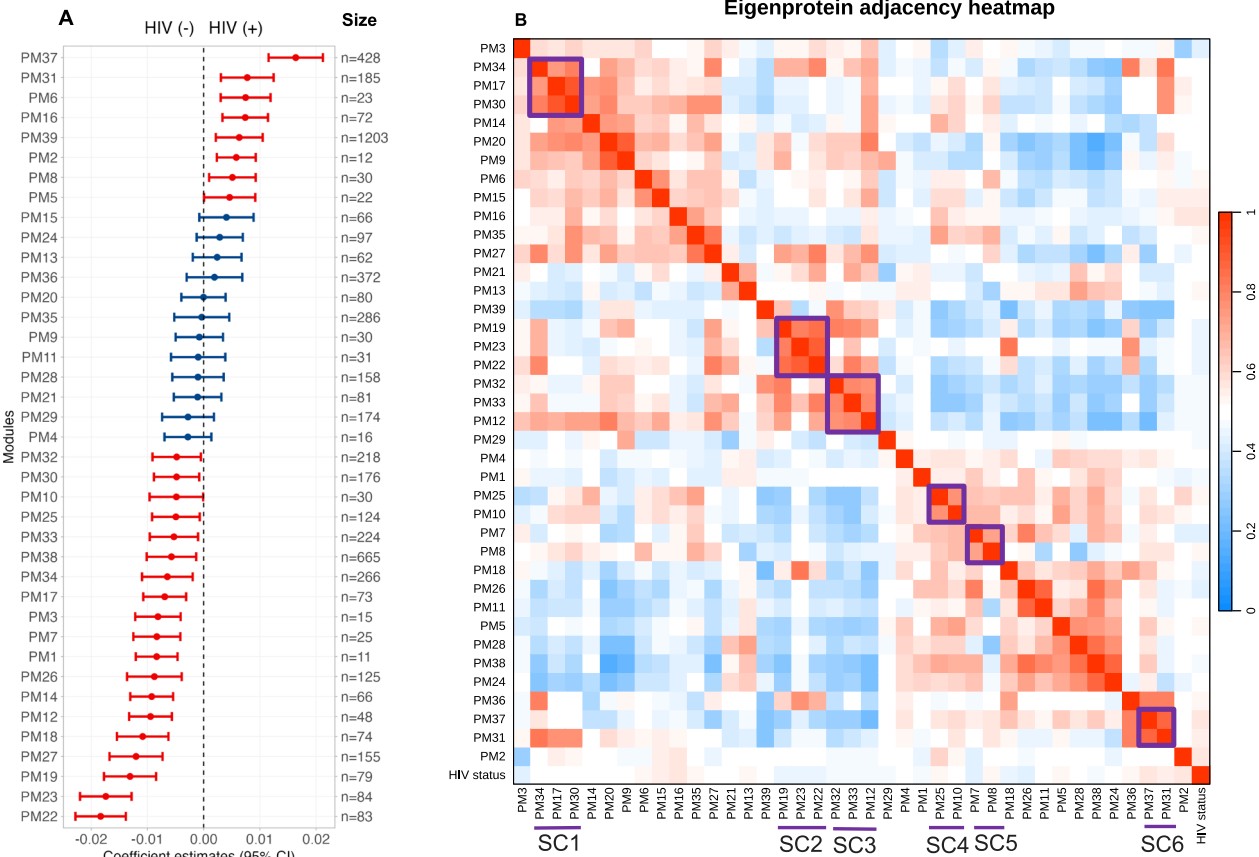

**Fig. 2 | Association of plasma proteome modules with HIV status among 689 children, including 79 with HIV. A** Forest plot of coefficient estimates showing association of plasma proteome modules measured at hospital discharge with HIV status. Estimates on the x-axis represent the beta-coefficients of this association. Points (centre of the bars) indicate beta coefficient estimates for every unit increase in plasma protein concentration in each module while error bars indicate 95% confidence interval. The red colour indicates significantly differentially expressed protein modules by HIV status. The size indicates the number of proteins in each module. **B** Eigenprotein adjacency heatmap showing the correlation between modules and HIV status, and superclusters. Purple coloured rectangle highlight HIV-related superclusters with eigenprotein correlations of ≥0.5. Superclusters only arising from HIV-related protein modules are highlighted. Abbreviations: PM, protein module, SC1 supercluster 1 comprised of modules 34, 17, and 30, SC2 supercluster 2 comprised of modules 19, 23 and 22, SC3 supercluster 3 comprised of modules 32, 33, and 12, SC4 supercluster 4 comprised of modules 25 and 10, SC5 supercluster 5 comprised of modules 7 and 8, SC6 supercluster 6 comprised of modules 37 and 31.

complement activation, humoral immune responses, and peptide transport, while blood coagulation, insulin-like growth factor receptor signalling, cellular and bone morphogenesis were decreased. Our results additionally indicate that among severely wasted children, HIV indirectly influences post-discharge growth by modulating biological processes such as insulin-like growth factor receptor signalling, bone morphogenesis, and humoral immune responses. Overall, irrespective of the HIV status, the study children experienced worsening linear growth trajectories post-discharge.

HIV is associated with increased risk of growth failure during the first years of life, and earlier HIV infection is linked with poorer growth especially in children who have not started antiretroviral treatment (ART). In ART naïve children living with HIV in Zimbabwe, the odds of stunting (HAZ: < −2) and wasting (WHZ: < −2) by 2 years of age was over 7-fold high for children with HIV compared to those without HIV[4]. In Zimbabwean and Zambian children treated for complicated severe malnutrition, HIV was associated with suboptimal nutritional recovery during 1-year follow up period[5]. Children with HIV usually improve in weight and height after initiating ART. In West Africa, children with HIV exhibited rapid increase in WAZ and WHZ in the first year of ART while HAZ increased gradually over the entire 5-year study period[24]. Similar findings have been reported in other African studies[25–27]. Our findings correspond with data from Africa that shows that more severely malnourished children with HIV have lower anthropometry at baseline and

experience greater catch-up growth over time, but do not reach community norms[26,28–30]. Except for HAZ, children with HIV showed greater post-discharge gains in MUAC, WAZ and WHZ, but these gains were insufficient for them to normalise to the levels of their counterparts. It is likely that the greater gains observed in the post-discharge period were partly due to therapeutic feeding which are rich in protein and energy and linked to faster catch-up growth[31–33]. Additionally, these gains could also be in part attributed to ART initiation given that 74% of the children with HIV in this study were ART-naïve at admission. The World Health Organisation (WHO) recommends immediate ART initiation in ART-naïve children with HIV following nutritional stabilisation[32]. Our findings, consistent with previous reports[34], showed a high rate of stunting in this cohort and HAZ remained unchanged over the 6 months post-discharge, irrespective of HIV status.

Children with HIV often suffer from opportunistic infections including tuberculosis, diarrhoea and prolonged malabsorption, with inflammation and enteropathy, potentially affecting weight and height gains[35]. A recent study among older Ethiopian children with HIV on ART reported a high incidence of opportunistic infections[36]. A clinical trial in four sub-Saharan African countries among ART-naïve adults and older children starting ART reported lower rates of tuberculosis, cryptococcal infection, oral/oesophageal candidiasis, and mortality in those receiving enhanced antimicrobial prophylaxis (comprising

**Table 2 | Significantly over-represented biological pathways**

| Modules | Size | Over-represented biological pathways | Bonferroni p-value | Databases |
|---|---|---|---|---|
| Modules positively associated with HIV | | | | |
| PM8 | 30 | Complement activation | 0.0001 | DAVID/WebGestalt |
| | | Complement activation[a] | 0.0008 | DAVID/STRING |
| PM37 | 428 | Regulation of type immune responses[a] | 0.04 | STRING |
| PM39 | 1203 | Translational initiation | 0.0001 | DAVID |
| | | Establishment of cellular localization | 0.0001 | DAVID |
| | | Early to late endosome transport regulation[a] | 0.01 | DAVID |
| | | Peptide transport | 0.0001 | WebGestalt |
| SC6-PM31,37 | 613 | Humoral immune response[a] | 0.005 | DAVID/STRING |
| Modules negatively associated with HIV | | | | |
| PM38 | 665 | Synaptic membrane adhesion | 0. 0001 | DAVID/STRING |
| | | Synaptic membrane adhesion[a] | 0. 0001 | DAVID |
| | | Axonogenesis | 0. 0001 | WebGestalt |
| PM1 | 11 | Blood coagulation | 0.004 | DAVID/WebGestalt |
| | | Regulation of blood coagulation[a]/regulation of fibrinolysis[a] | 0.001 | DAVID/STRING |
| PM12 | 48 | Mitotic cell cycle phase transition | 0.04 | WebGestalt |
| PM32 | 218 | Purine nucleotide biosynthetic process | 0.001 | DAVID/STRING |
| | | 5-phosphoribose 1-diphosphate biosynthetic process[a] | 0.03 | DAVID |
| | | 5-phosphoribose 1-diphosphate biosynthetic process | 0.01 | WebGestalt |
| PM30 | 176 | Innate immune response | 0.04 | WebGestalt/STRING |
| PM33 | 224 | Activation of innate immune response[a] | 0.0001 | DAVID |
| | | Ribosomal small subunit biogenesis | 0.01 | DAVID/STRING |
| | | mRNA processing | 0.01 | WebGestalt |
| PM34 | 266 | Défense response to other organism | 0.02 | STRING |
| PM7 | 25 | Insulin-like growth factor receptor signalling | 0.01 | DAVID/WebGestalt |
| | | Positive regulation of insulin-like growth factor receptor signalling[a] | 0.03 | DAVID/STRING |
| PM19 | 79 | Positive regulation of cell proliferation[a] | 0.0001 | DAVID |
| | | Hindlimb morphogenesis | 0.01 | WebGestalt/STRING |
| PM26 | 125 | Collagen fibril organisation | 0.0001 | DAVID |
| | | Bone mineralisation[a] | 0. 0001 | DAVID |
| | | Bone morphogenesis | 0. 0001 | WebGestalt |
| PM22 | 83 | Telomere organisation | 0.0001 | DAVID |
| | | Regulation of gene expression, epigenetic[a] | 0. 0001 | DAVID |
| | | Chromatin assembly | 0.007 | WebGestalt |
| PM23 | 84 | Regulation of signalling receptor activity | 0. 0001 | WebGestalt |
| | | Positive regulation of MAPK cascade[a] | 0.003 | DAVID |
| PM27 | 155 | Positive regulation of protein phosphorylation | 0.03 | STRING |
| Superclusters of HIV-related modules | | | | |
| SC1-PM17,30,34 | 513 | Inflammatory response[a] | 0.027 | DAVID/STRING |
| | | Innate immune response[a] | 0.017 | DAVID |
| SC2-PM19,22,23 | 319 | T-helper 2 cell cytokine production | 0.006 | DAVID |
| | | T-helper 2 cell cytokine production[a] | 0.006 | DAVID |
| | | Regulation of signalling receptor activity | 0.001 | WebGestalt |
| SC3-PM12,32,33 | 489 | Ribosomal small subunit biogenesis | 0. 0001 | DAVID |
| | | Regulation of intrinsic apoptotic signalling by p53 mediator[a] | 0.04 | DAVID |
| | | mRNA metabolic process | 0.0001 | WebGestalt |
| Supercluster containing both negatively and positively HIV-related modules | | | | |
| SC5-PM7,8 | 55 | Complement activation | 0.001 | DAVID |
| | | Cytolysis[a] | 0.01 | DAVID |
| | | Regulation of humoral immune response | 0.01 | WebGestalt |

[a]Homo sapiens used as the reference background for calculating fold enrichment for Gene Ontology enrichment analysis for biological processes. Enrichment was assessed with Fisher's exact or hypergeometric tests, and P value adjusted for Bonferroni correction.

isoniazid, fluconazole, azithromycin, and single-dose albendazole) compared to the standard-prophylaxis group[37]. Additionally, patients receiving ready-to-use supplementary food (RUSF) had greater weight, body mass index, and MUAC gain compared to the non-RUSF group but without improvement in mortality or clinical outcomes[38]. Taken together, in the era of ART, it could be possible that achieving sufficient growth recovery among severely malnourished children with HIV requires more than ART alone.

HIV exacerbates inflammation in children with underlying SM[18]. Previously, we found a negative association between systemic inflammation and early post-discharge growth in severely malnourished children without HIV following acute illness[39]. Persistent humoral and cellular immune activation is one of the hallmarks of HIV infection[40] often associated with hypergammaglobulinemia (elevated levels of immunoglobulins)[41]. Our current findings are consistent with this, revealing that HIV was positively associated with humoral immune responses which were negatively associated with baseline MUAC and with 90-day post-discharge WAZ and WHZ. Additionally, IL-6 and TNFα associated with exacerbated inflammation in children with HIV-SM compared to SM alone[18] controlled the network connectivity of the

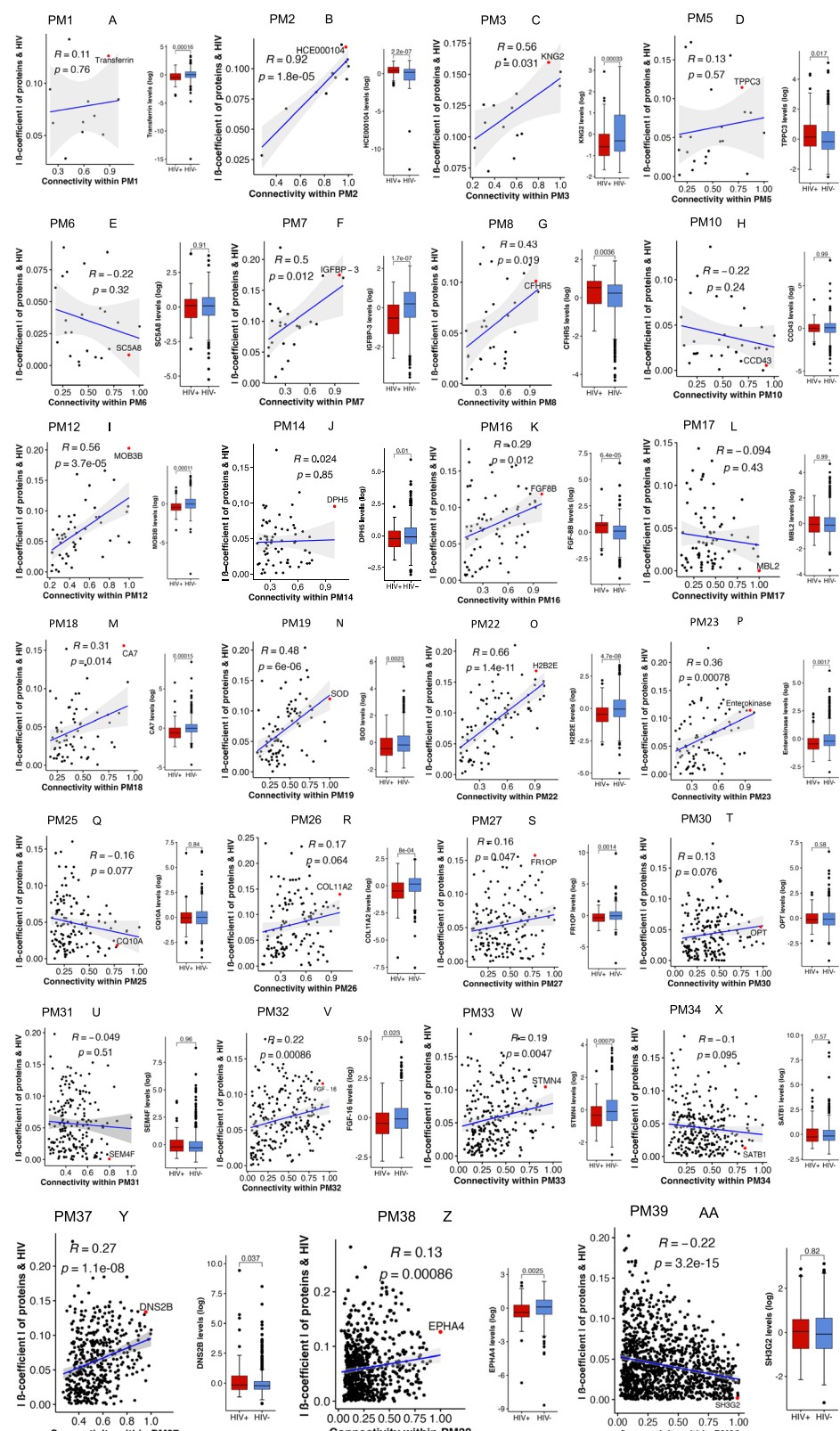

module associated with humoral immune responses. Although increased complement activation pathways in children with HIV were not associated with post-discharge growth, their elevation alongside heightened humoral immune responses may suggest ongoing active infection or gut-systemic microbial product translocation which could increase metabolic demands and divestment in nutrient utilisation thereby affecting growth. In children with SM, elevated inflammatory markers have been associated with reduced growth factor levels[18]. HIV infection is known to be associated with decreased levels of IGF-I, IGF-II and IGFBP-3[42,43]. We observed a negative association between HIV and insulin-like growth factor receptor signalling pathway which was in turn positively associated with baseline WAZ. Moreover, children with HIV exhibited lower levels of IGF-I, IGFBP-3, IGFBP-5, GHR, acid labile subunits (ALS) and leptin. IGF-I is a central hormone involved in

**Fig. 3 | Identification of hub proteins in modules associated with HIV. A–AA** Scatter plots showing correlation of absolute effect sizes of protein-HIV significance (y-axis) and intra-modular connectivity (x-axis) among modules significantly associated with HIV. Positive correlations were observed in 15 protein modules while PM39 displayed a negative correlation. Modules PM1, PM5, PM6, PM10, PM14, PM17, PM25, PM26, PM30, PM31 and PM34 did not show significant correlations. Correlation coefficients (R) with their corresponding significance are shown where $p < 0.05$ were considered significant. The blue regression line shows this relationship while the error bands represent the 95% confidence interval for the regression line. Hub proteins are highlighted by red dots and labelled accordingly.

Box plots show the expression levels of hub proteins by HIV status (children with HIV, $n = 79$; children without HIV, $n = 610$). Data are presented as median values with interquartile ranges (IQR). A two-sided t-test was used to compare expression levels, with Bonferroni-adjusted $p < 0.05$ considered statistically significant. Box plots indicate; median (middle line); 25th (first quartile, Q1) and 75th (third quartile, Q3) percentile (box limits); error bars (whiskers) represent 1.5*Q1 and Q3 while single points outside the error bars represent outliers. Abbreviations: PM, protein module; |β-coefficient|, absolute beta coefficient estimate of a relationship between proteins in a module and HIV – the protein-HIV significance; R, correlation coefficient; p, p-value.

**Table 3 | Characteristics of the participants with plasma proteome data at discharge, who were included in the HIV mechanistic analysis**

| Characteristics at discharge | | HIV⁻ (n = 179) | HIV⁺ (n = 38) | | Total (n = 217) |
|---|---|---|---|---|---|
| **Demographic** | | | | | |
| Sex – Males N (%) | | 102 (57%) | 20 (53%) | | 122 (56%) |
| Age, months – Median (IQR) | | 13.9 (8.9–17.8) | 14.9 (7.3–19.3) | | 13.9 (8.9–17.9) |
| **Site** | | | | | |
| Banfora | | 35 (20%) | 2 (5.3%) | | 37 (17%) |
| Blantyre | | 15 (8.4%) | 5 (13%) | | 20 (9.2%) |
| Kampala | | 63 (35%) | 9 (24%) | | 72 (33%) |
| Kilifi | | 20 (11%) | 8 (21%) | | 28 (13%) |
| Migori | | 28 (16%) | 12 (32%) | | 40 (18%) |
| Nairobi | | 18 (10%) | 2 (5.3%) | | 20 (9.2%) |
| **Anthropometry** | | | | | |
| MUAC (cm) | Median (IQR) | 10.80 (10.15–11.30) | 10.20 (9.58–11.06) | | 10.65 (10.00–11.30) |
| | Mean (SD) | 10.60 (±0.932) | 10.20 (±1.10) | | 10.5 (±0.975) |
| WAZ score | Median (IQR) | −3.96 (−4.67 to −3.18) | −4.52 (−4.98 to −3.44) | | −4.01 (−4.79 to −3.19) |
| | Mean (SD) | −3.93 (±1.22) | −4.27 (±1.25) | | −3.99 (±1.23) |
| WHZ score | Median (IQR) | −2.95 (−3.84 to −2.20) | −3.17 (−4.24 to −2.52) | | −3.01 (−3.91 to −2.22) |
| | Mean (SD) | −3.93 (±1.22) | -4.27 (±1.25) | | −3.02 (±1.30) |
| HAZ score | Median (IQR) | −3.35 (−4.39 to −2.39) | −3.63 (−4.59 to −2.10) | | −3.39 (−4.46 to −2.34) |
| | Mean (SD) | −3.32 (±1.68) | −3.33 (±1.84) | | −3.32 (±1.70) |
| Oedema – Yes | | 15 (8.4%) | 2 (5.3%) | | 17 (7.8%) |
| **Clinical illness at admission – N (%)** | | | | | |
| Diarrhoea | | 79 (44%) | 22 (58%) | | 101 (47%) |
| Pneumonia | | 84 (47%) | 15 (39%) | | 99 (46%) |
| Malaria Positive (RDT) | | 28 (16%) | 4 (11%) | | 32 (15%) |
| Sepsis | | 5 (2.8%) | 2 (5.3%) | | 7 (3.2%) |
| **On Co-trimoxazole prophylaxis and Antiretroviral treatment at admission – N (%)** | | | | | |
| Co-trimoxazole prophylaxis | | 2 (1.1%) | 15 (39%) | 17 (7.8%) | |
| Antiretroviral treatment (ART) | | – | 9 (24%) | – | |

Data are median (IQR), mean (SD) or count, n (%).

*RDT* rapid diagnostic test, *TB* tuberculosis, *MUAC* mid-upper arm circumference, *WAZ* weight-for-age, *WHZ* weight-for-height, *HAZ* height-for-age, *IQR* interquartile range, *SD* Standard deviation.

growth, metabolism and tissue repair, mostly occurring in ternary complexes with IGFBP-3 and ALS in the systemic circulation, which prolong its half-life[44]. Taken together, elevated humoral immune responses and inflammation may contribute to decreased growth factor signalling in children with HIV, potentially impacting growth. Given leptin's role in suppressing appetite, regulation of energy expenditure and immune function, and whose circulating levels are positively correlated with body mass index and adipose tissue mass in mammals[45], Lower leptin levels at discharge in children with HIV-SM suggest promotion of orexigenic signalling to promote nutrient intake and weight recovery. Low leptin levels were reported in Ugandan and Kenyan children with SM who died during and after hospitalisation respectively[31,46].

Children with HIV have reported deficits in several structural bone parameters including bone mineral density and content compared to those without HIV[47,48]. We found a negative association between HIV and biological process involved in bone mineralisation which in turn exhibited a negative association with baseline WAZ and 90-day post-discharge MUAC. Bone mass loss in individuals with HIV is influenced by various factors including viral load, ART, nutritional deficiencies, hormonal and immune dysregulation[49]. In ART-naïve adults with HIV, upregulation of B cell expression of receptor activator of nuclear factor kappa beta ligand (RANKL) was linked to osteoclastic bone loss[50]. Consistently, in HIV-1 transgenic rat models, reduced bone mineral density and structure was due to increased RANKL levels[51,52]. RANKL, also known as tumour necrosis factor ligand superfamily member 11 (TNFSF11) is a member of the TNF superfamily and is involved in osteoclastogenesis[53], the process by which osteoclasts are formed. In the current study, children with HIV compared to those without HIV also exhibited elevated humoral

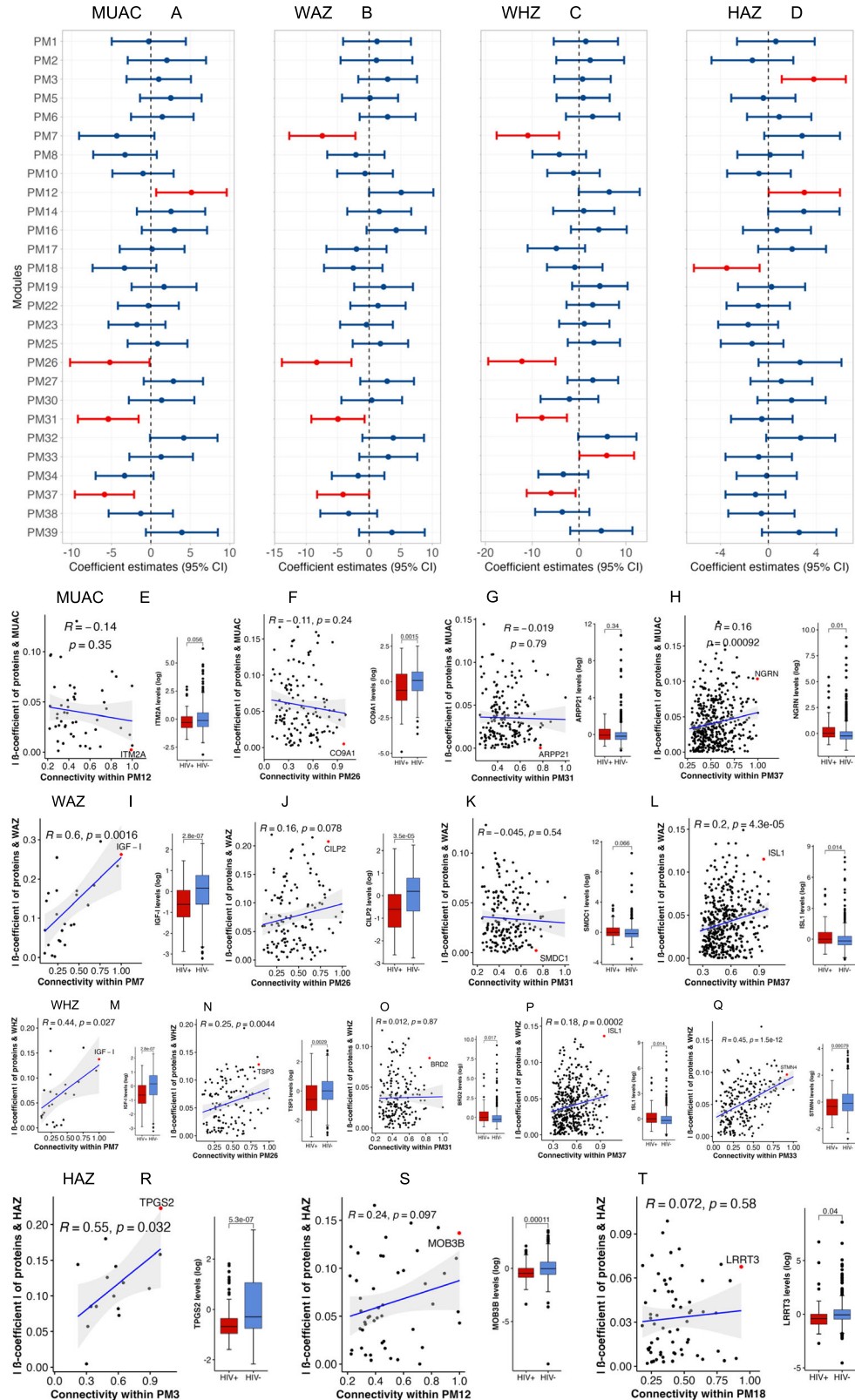

immune responses and TNFSF11 levels, alongside decreased levels of IGF-I, the major hormone required for postnatal bone elongation[54]. Additionally, children with HIV exhibited decreased levels of thrombospondin-3 and 4; proteins involved in the development of the bone and skeletal muscles[55]. Moderately acutely malnourished children receiving microbiota-directed complementary food exhibited a significant increase in weight-for-length z scores which correlated with higher plasma thrombospondin-4 levels compared to those who received RUSF[56]. Taken together, these findings suggest a potential explanation for the decreased bone mineralisation in children with HIV in this study. Reduced bone mineralisation might reflect a shift from height growth to favour muscle and fat accretion.

**Fig. 4 | Association of HIV infection-related protein modules with 90-day post-discharge growth and hub proteins identification.** Analysis included 217 children of whom 38 had HIV infection. **A**–**D** Forest plots showing association of HIV-related protein modules at discharge with 90-day post-discharge MUAC, WAZ, WHZ, and HAZ. Estimates on the x-axis represent the beta-coefficients of this association. Points (centre of the bars) indicate beta coefficient estimates for every unit increase in plasma protein concentration in a given module while error bars represent 95% confidence intervals. Red colour indicates estimate of modules significantly associated with 90-day post-discharge anthropometric measurements. **E**–**T** Scatter plots illustrating the correlation between absolute effect sizes of protein-growth significance and intra-modular connectivity as shown by the blue regression line. The association coefficient, R, and significance of association, P, are indicated accordingly. The error bands represent the 95% confidence interval for the regression line. Red dots on the plots represent hub proteins in each module. Box

plots show the expression levels of hub proteins by HIV status (children with HIV, $n = 38$; children without HIV, $n = 79$). Data are presented as median values with interquartile range (IQR). A two-sided t-test was used to compare expression levels of hub proteins between children with and without HIV, denoted by red and blue colours, respectively. *P* values were adjusted for multiple testing using the Bonferroni correction criterion. Box plots indicate; median (middle line); 25th (first quartile, Q1) and 75th (third quartile, Q3) percentile (box limits); error bars (whiskers) represent 1.5*Q1 and Q3 while single points outside the error bars represent outliers. Abbreviations: MUAC mid-upper arm circumference, WAZ weight-for-age z score, WHZ weight-for-height z score, HAZ height-for-age z score, PM protein module, |β-coefficient| absolute beta coefficient estimate of a relationship between proteins in a module and 90-day anthropometric measurements, R correlation coefficient, p p-value where $p < 0.05$ was considered significant.

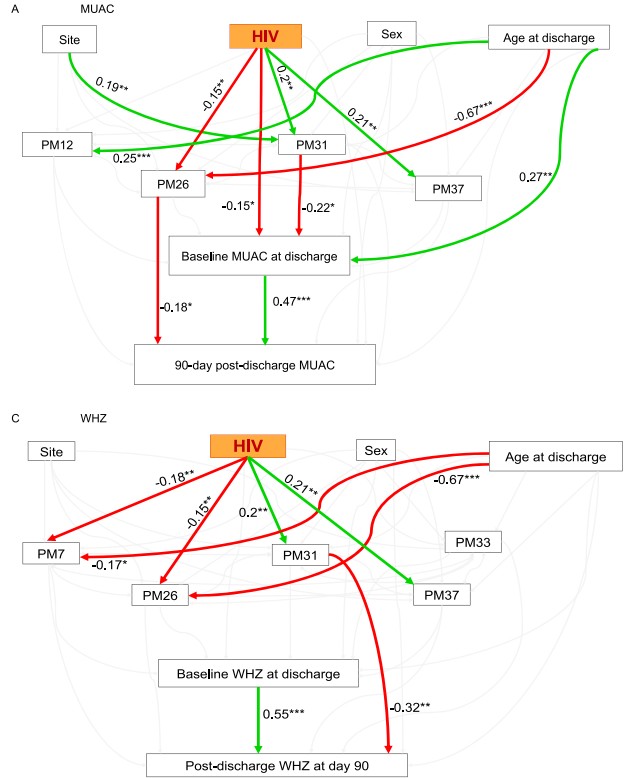

**Fig. 5 | Path diagrams depicting influences of HIV on post-discharge anthropometry. A** MUAC SEM framework. **B** WAZ SEM framework. **C** WHZ SEM framework. **D** HAZ SEM framework. The SEMs incorporated HIV status, HIV infection-related protein modules associated with 90-day post-discharge anthropometry, post-discharge anthropometric measurements, baseline anthropometry at discharge, site, sex, and age at discharge. Green and red coloured lines depict significant positive and negative associations, respectively, with asterisk indicating significant coefficient estimates. Grey coloured lines show non-significant associations. Singe-headed arrows depicts a path linking exogenous to endogenous variable while double-headed arrows represent covariance between protein modules. Path coefficients are standardised estimates of individual relationship within

the structural equation model resulting from a simple linear regression. The overall model fit was assessed using the chi-square test and supplemented by additional fit indices; comparative fit index (CFI), root mean square error for approximation (RMSEA) and standardised root mean squared residual (SRMR) to confirm model adequacy, see the method section. Abbreviations: HIV, Human immunodeficiency virus – the primary exogenous variable of the structural equation framework; SEMs, structural equation models; MUAC, mid-upper arm circumference; WAZ, weight-for-age z score; WHZ, weight-for-height z score; HAZ, height-for-age z score; PM, protein module; site, enrolment site; and sex defined by biological attribute. The asterisks indicate significance gradient: *; p-value less than or equal to 0.01; **, *p*-value less than or equal to 0.001; and ***, p-value less than or equal to 0.0001.

Combination ART is recommended for all children with HIV upon diagnosis and is effective in controlling viral replication and disease progression. Paradoxically, bone loss during ART initiation may be greater than that estimated to result solely from HIV infection, with decline in bone mineral density estimated to occur in the first 2 years of ART initiation[57]. Calcium and vitamin D supplementation at ART initiation in ART-naïve adults with HIV attenuated bone loss[58]. Therefore, such interventions alongside early ART initiation in children with HIV should be considered, particularly in cases of low calcium dietary intake and low vitamin levels.

Protein glutamylation, a post-translational modification process in which glutamate residues are added to the target protein is required for regulation of cellular processes including stabilisation of the cytoskeleton, especially microtubules. A recent review has highlighted the role of post-translational modification and stabilisation of microtubules in regulation of intracellular trafficking, viral assembly and release during HIV infection[59]. Glutamylation of HIV-1 protein p6 has been proposed as one strategy by which HIV-1 evades host immune response[60]. Our results show that HIV was negatively associated with biological processes related to polyglutamylation which subsequently

showed positive relationship with 90-day post-discharge HAZ suggesting a potential role of glutamylation in linear growth. Glutamate supplementation has been associated with improved foetal growth in sows and reduced preweaning mortality in piglets[61].

Adequate nutrition is vital for synaptic development and overall growth, including height gain[62]. Malnutrition can adversely affect synaptic function, cellular morphogenesis and growth processes. Our results revealed that biological processes related to cellular morphogenesis and postsynaptic membrane assembly were negatively associated with baseline and 90-day post-discharge HAZ respectively, irrespective of HIV status. This suggests their potential involvement in linear growth among severely malnourished children independent of HIV.

Other than biological mechanisms, growth recovery in young children following acute illness has been associated with factors including adverse caregiver characteristics (caregiver education, mental health, illness and employment status), household socio-economic status and access to health care[63,64]. While socio-demographic factors are important, they likely operate through biological processes as demonstrated in this study. Therefore, the mechanisms enumerated here could be more proximal to growth than upstream socio-demographic factors.

Future work should investigate the longitudinal normalisation of biological processes and their relationship with nutritional recovery during convalescence. Additionally, understanding the effect of HIV in well-nourished community children could help establish community norms associated with recovery and evaluate interventions to improve the biological processes identified in this study. The study's strengths lie in the inclusion of children from 6 study sites in 4 African countries enhancing the generalisability of the results to a wider population within resource-constrained settings. Additionally, the use of SEM to identify potential modifiable pathways through which HIV influences post-discharge growth among severely wasted children. This addresses a gap in understanding the interaction between HIV and SM in children. Limitations include incomplete data on ART use, with the study assuming management based on 2013 WHO guidelines[32]. Additionally, there were no data on CD4+ counts and viral loads to assess the severity of HIV infection. Furthermore, the SEM frameworks did not include variables such as household socio-economic status, which may potentially affect the observed relationships. Lastly, the requirement of the study to assess anthropometric measurements over time meant that children who died during follow up were excluded, potentially introducing survivor bias.

In conclusion, HIV appears to modulate early post-discharge growth during convalescence by influencing systemic biological processes such as those linked to bone mineralisation, IGF-I signalling and humoral immune responses. These findings suggest a need for clinical trials to investigate interventions that target these biological processes (such as calcium and vitamin D supplementation) along with early ART initiation in children with HIV-SM.

## Methods

### Study design and population

This study is nested within the Childhood Acute Illness and Nutrition (CHAIN) Network cohort study that aimed to characterise the biomedical and social risk factors for mortality in acutely ill young children[65]. Briefly, the CHAIN cohort enroled children aged 2–23 months during admission to one of nine hospitals in six countries in sub-Saharan Africa and South Asia. Children were admitted to hospital with acute illness from malaria, diarrhoea, pneumonia, sepsis, systemic inflammatory response syndrome (SIRS), HIV infection, anaemia, and comorbidities including severe malnutrition. Acutely-ill children were enroled and classified into three nutritional strata based on the mid-upper arm circumference (MUAC); not wasted (MUAC $\geq 12.5$ cm for age $\geq 6$ months or MUAC $\geq 12.0$ cm for age <6 months), moderately

wasted (MUAC 11.5 cm to <12.5 cm for age $\geq 6$ months or MUAC 11.0 cm to <12.0 cm for age <6 months), and severely wasted or kwashiorkor (MUAC < 11.5 cm for age $\geq 6$ months or MUAC < 11.0 cm for age <6 months or bilateral pedal oedema unexplained by other medical causes)[65]. Children were followed up for 6 months after discharge from hospital with planned visits at days 45, 90 and 180. A total of 3101 children were enroled between November 20, 2016 and January 31, 2019 of whom 5.1% ($n = 157$) had HIV infection, defined by polymerase chain reaction (PCR) or antigen testing per national protocols. CHAIN collected detailed clinical, demographic, anthropometric, laboratory and social exposures as well as blood, stool and rectal swabs as described previously[65].

The current study is a case-control analysis, focussing on the effect of HIV infection on post-discharge growth following acute illness and explores how HIV infection modulates systemic biological mechanisms that influence post-discharge growth among severely wasted children hospitalised with acute illness. We initially categorised all children with HIV infection ($n = 157$) as cases, while controls consisted of children without HIV, 24% of whom were selected using a random number generator. To focus specifically on post-discharge growth, we excluded children who died during their hospital stay ($n = 182$, including 45 children with HIV) and those from Asian study sites where HIV prevalence is very low. Consequently, our HIV comparative analysis (examined the effect of HIV on six months post-discharge growth) included a total of 112 children with HIV infection (cases) and 722 children without HIV (controls), as illustrated in Fig. 1A. The comparison of systemic proteome signatures by HIV status analysed 689 children of whom 79 had HIV. For the mechanistic analysis investigating influence of HIV on post-discharge growth, only severely wasted children ($n = 217$ of whom 38 children had HIV) at discharge who survived up to 3 months and had proteome data were included.

### Ethics

Ethical approvals were obtained from every participating site or collaborating institutions and from the University of Oxford. All caregivers provided written informed consent for their children to participate in the study. The study protocol was reviewed and approved by the Oxford Tropical Research Ethics Committee, United Kingdom; Scientific and Ethical Review Unit (SERU), Kenya Medical Research Institute, Kenya; Makerere University School of Biomedical Sciences Research Ethics Committee and The Uganda National Council for Science and Technology, Uganda; The University of Malawi and COMREC, Kamuzu University of Health Sciences, Malawi; The University of Ouagadougou and Comité d'éthique institutionnel du Centre MURAZ, Burkina Faso.

### Plasma proteomics and data pre-processing

Proteins in plasma samples obtained from children at discharge from the hospital were quantified using the aptamer-based SomaScan™ assay[66,67]. The SomaScan™ assay employs Slow-Off rate Modified Aptamers (SOMAmers) to profile plasma proteome. It quantitatively transforms protein epitopes into SOMAmer-based DNA signals, subsequently measured via DNA-hybridisation microarrays. The resultant text-based ADAT files were imported, transformed and annotated using a free and open-source R package called readat[68]. Subsequently, the non-human protein targets were filtered retaining only proteins from humans ($n = 7335$ proteins) which were log-transformed, scaled to unit variance by autoscaling and mean-centred. Data provided from the SomaScan™ assay did not contain missing values. The distribution of raw intensities for each protein was visualised through histograms, boxplots and Principal Component Analysis (PCA) plots, which can be accessed via this link https://mudiboevans.shinyapps.io/shinyapp/. The box and PCA plots are stratified by sex (biological attribute), site of enrolment, nutritional (no wasting, moderate wasting and severe wasting) and HIV status.

## Data analysis

**Baseline characteristics.** Participant characteristics including clinical, demographic (sex, age, site), nutritional status and anthropometry at hospital discharge were summarised using median with interquartile ranges or mean with standard deviations if continuous and proportions if categorical, stratified by HIV status.

**Growth from discharge through 6 months of post-discharge follow up.** We compared growth between children with and without HIV by examining mid-upper arm circumference (MUAC), weight-for-age z score (WAZ), weight-for-height z score (WHZ) and height-for-age z score (HAZ) from discharge to 180 days post-discharge. Fixed-effects panel model specification was used with individual effects included in the model as unobserved time-invariant characteristics of the child at a given time point. The model included an interaction term of HIV status with time point and adjusted for sex, recruitment site and baseline anthropometry at discharge for each growth parameter as shown in Eq. 1.

$$Y_{it} = \alpha_i + \text{Time}_t + \text{baseline anthropometry} + \text{sex} + \text{site} + \text{Time}_t : \text{HIV status} + u_{it}$$ (1)

where $Y_{it}$ is either MUAC, WAZ, WHZ or HAZ at time point $t$; $\alpha_i$ (i = 1…n) is the individual fixed effect; $\text{Time}_t$ is the time trend variable $t$; baseline anthropometry is the growth parameter at discharge while sex and site are time-invariant covariates; $\text{Time}_t$:HIV status is the interaction term of time with HIV status at a given time point and $u_{it}$ is the error term. Children without HIV were used as the reference group in the interaction part of the model. This was implemented in R using the plm[69] package (version 2.6-4).

Given that the parent CHAIN study ensured accuracy by double measuring and verifying anthropometric data in hospitalised children, no outlier values were excluded from the analysis.

**Construction of the protein correlation network.** We used a weighted correlation network analysis, which assumes that linkages within biological networks have a non-random scale-free topology[70]. In biological settings, a networks topography is considered to have reached near scale-free topography at $r^2 \geq 0.8$. Using the weighted gene correlation network analysis (WGCNA)[71] package (version 1.73) in R, we reduced the dimensionality of the plasma proteomics dataset into clusters of tightly correlated proteins called modules. Firstly, to achieve a scale-free topology criterion, we generated sets of soft thresholding powers ranging from 1 to 50 (Supplementary Fig. 6A). Optimal soft thresholding power of 9 with $r^2 = 0.85$ and minimum mean connectivity was chosen. Secondly, a biweight mid-correlation, $s_{ij} = \text{bicor}(x_i, x_j)$ matrix was generated between every pair of proteins ($i$ and $j$). This correlation matrix was then transformed into an adjacency matrix, $a_{ij} = |s_{ij}|^\beta$ through power transformation using the soft thresholding power of 9. Through power transformation, weak and negative correlations are punished while strong correlations are amplified, this in turn helps to augment signal-noise ratio in the adjacency matrix thus increasing robustness of the network. Among various potent soft thresholding powers, 9 produced a network with fewer unclassified proteins, making it the optimal choice for analysis. For better biological interpretation, a signed network of proteins was constructed. By applying hierarchical clustering algorithm implemented in the WGCNA[71] package, tightly correlated proteins were clustered into modules using the one-step automated blockwiseModules function. The minimum number of proteins forming a module was set to 10. Other parameters used for network construction include: maxBlockSize of 20,000 to ensure that the entire proteome is analysed as a single block and not as small blocks. The parameter for merging the modules was set at 0.25.

**Association between HIV status and protein modules at discharge.** Each protein member of a module is characterised by an eigenprotein, $E^{(q)}$, which is the first principal component of a given module obtained through singular value decomposition. The module eigenprotein primarily represents the collective behaviour of that specific module. The association between individual protein modules, treated as the outcome variable, and HIV status was implemented using multivariate linear regression with inverse probability weighting (IPW) to balance the baseline characteristics between children with and without HIV[72,73]. Weight for each observation selected into the nested case-control study was generated by computing the probability of a child being HIV positive given their age, sex, nutritional status, clinical syndromes at admission to hospital (malaria, diarrhoea, pneumonia) and site of recruitment. The R script for computing weights using IPW is archived at the Harvard Dataverse website[74] under https://doi.org/10.7910/DVN/D8HZLJ while Supplementary Eqs. 1–3 illustrating IPW are found in Supplementary Methods. The association between HIV status and protein modules was executed using *lmer* function in the lme4[75] R package (version 1.1-35.5) as shown in Eq. 2.

$$\text{ME}_i^{(q)} = \beta_0 + \beta_i \text{HIV status} + (1|\text{site}) + u; \text{weighted}$$ (2)

where $\text{ME}_i^{(q)}$ is the eigenprotein for the respective protein module; $\beta_0$ is the y-intercept; $\beta_i$ is the beta-coefficient of the association between HIV status and eigenprotein for module i; HIV status coded as 1 or 0 represented children with and without HIV respectively; site is the enrolment site here modelled as a random effect and $u$ is the error term. Weighted means the model used IPW as described above.

Significant associations between protein modules and HIV status were determined at $p < 0.05$ after adjusting for multiple comparison for the 39 identified modules using Bonferroni correction.

**Biological pathway analysis.** To comprehensively uncover biological processes represented within each differentially expressed protein module, we assessed Gene Ontology (GO) enriched biological processes using online versions of The Database for Annotation, Visualisation and Integrated Discovery (DAVID)[76] v2023q4 release, WEB-based Gene SeT AnaLysis Toolkit (WebGestalt)[77] 2019 version and the STRING database[78] version 12.0. STRING is a public database of known and predicted protein-protein interactions. The interactions include direct (physical) and indirect (functional) associations stemming from computational prediction, from knowledge transfer between organisms and from interactions aggregated from other (primary) databases. These three databases were used independently to explore biological processes. Homo sapiens and list of human proteins ($n = 7335$) from the SomaScan™ platform served as the background for calculating fold enrichment. Significance of GO-enriched biological processes was determined based on a Bonferroni corrected p-value of <0.05 and fold enrichment score. Protein connectivity within the modules was visualised in Cytoscape[79] version 3.10.2. Proteins in the network that occurred as singletons (not connected to each other) were removed from the final network output.

**Hub proteins identification in modules associated with HIV status.** As databases may not be able to ascertain all biological pathways involved in each module, we further identified hub proteins, which aided in understanding the biological mechanisms underlying the modules. To identify hub proteins, first we determined the association between expression profile of every protein in a module and HIV status to obtain protein-HIV significance, $P{:}\text{HIV}S_i = |\text{cor}(x_i, \text{HIV})|^\beta$, as an absolute value of the correlation between protein expression profiles in a module and HIV status. For each module, we further computed intra-modular connectivity which defines how proteins are connected within a given module. For ease of interpretation, connectivity values were scaled by dividing the within connectivity with maximum

connectivity for that module, $Ki = ki/kmax$. This was implemented using *intramodularConnectivity.fromExpr* function for a signed network within the WGCNA R package[71]. The absolute effect sizes of protein-HIV significance, $P$:HIV$S_i$, were then plotted against the scaled intra-modular connectivity, revealing hub proteins. The slope of the regression line is the hub protein significance, i.e., hub protein significance $= \sum_i P : HIVS_i K_i / \sum_i (K_i)^2$. Subsequently, the expression profiles of the hub proteins were compared between children with and without HIV, with p-values adjusted for multiple testing for the 27 modules associated with HIV status using Bonferroni correction.

**Association between HIV-related protein modules and 90-day post-discharge growth.** To examine the link between HIV and post-discharge growth among severely wasted children, we initially identified HIV-related protein modules associated with post-discharge anthropometry. Here, anthropometric measurements including MUAC, WAZ, WHZ and HAZ at 90 days after hospital discharge were used. Post-discharge growth was defined as anthropometric measures at day 90 after accounting for baseline anthropometry. To determine which HIV-related protein modules at discharge were associated with 90-day post-discharge growth, linear mixed-effects regression models were employed adjusting for anthropometry at discharge, age at discharge and sex, with site included as a random effect. This was implemented using *lmer* function in the lmertest[80] R package (version 3.1-3). Models were built for individual anthropometry, as shown in Eq. 3.0. We used 90-day post-discharge anthropometric measurements in this analysis because they are closer to the sample collection time and are more likely to be related to the biological processes measured at discharge, compared to the 180-day measurements.

$$\begin{aligned} Post-discharge\ anthropometry\ (at\ day\ 90) = \ &HIV\text{-}related\ ME_i^{(q)} \\ &+ anthropometry\ at\ discharge \\ &+ age\ at\ discharge + sex + (1|site) + u \end{aligned}$$
(3)

where post-discharge anthropometry (MUAC, WAZ, WHZ and HAZ) is the anthropometric measurement at day 90 after hospitalisation; HIV-related $ME_i^{(q)}$ is the eigenprotein of HIV-related protein module $i$ identified using Eq. 2.0 above; anthropometry at discharge is the baseline anthropometric indices taken upon hospital discharge; site is the random effect of the model and $u$ is the error term.

Candidate hub proteins of anthropometry-associated modules were determined by the correlation between the absolute effect size of protein-anthropometry significance and intra-modular connectivity.

**Pathways linking HIV, proteome and post-discharge anthropometry using structural equation modelling.** We examined pathway(s) linking HIV to 90-day post-discharge anthropometry using structural equation modelling (SEM). Simple linear regression models were built to assess the effect of HIV on 90-day post-discharge anthropometry based on our hypothesised base path model (Supplementary Fig. 5). The primary exogenous and endogenous variables in the SEMs were HIV status and 90-day post-discharge anthropometry, respectively, adjusting for baseline anthropometry. Other variables included HIV-related protein modules significantly associated with 90-day post-discharge anthropometry, baseline anthropometry at discharge, age at discharge, sex and site of recruitment. In the SEM models we allowed for covariance between protein modules. SEM models included children with missing 90-day post-discharge anthropometry thus, coefficients were estimated using full information maximum likelihood estimator (FIML), accounting for missing observations[81]. This was implemented in the lavaan[82] package (version 0.6-19) in R using the *sem* function. Standardised path coefficients were used to assess the relationship between exogenous and endogenous variables. Associations with significance levels <0.05 were considered important.

Model fits for the SEM were evaluated using the Chi-square ($\chi^2$) test statistic, with a $p > 0.05$ indicating no significant difference between the hypothesised and the fitted models. This implies that the conceptualised model potentially describes a true relationship under investigation in the general population. Other model fit metrics included the comparative fit index (CFI, where CFI > 0.90 indicates good model fit), root mean square error for approximation (RMSEA, <0.06 represents good fit, <0.09 reasonable fit) or standardised root mean squared residual (SRMR, <0.06 represents good fit)[83]. Supplementary Data 1 summarises model parameter estimates and fit measures. A general analysis workflow of the present study is shown in Supplementary Fig. 7.

### Reporting summary
Further information on research design is available in the Nature Portfolio Reporting Summary linked to this article.

## Data availability
The data that support the findings of this study is available at Havard Dataverse repository[74] through this link https://doi.org/10.7910/DVN/D8HZLJ. The data contain sensitive information about study participants and may include identifiers that could compromise confidentiality. To ensure participant privacy and compliance with ethical guidelines and data protection regulations, access to the data is restricted. Access to this data require submission of a formal request to the data governance committee via dgc@kemri-wellcome.org. Details on how to request the data and submission of a data request form are found on the Havard Dataverse website using this link https://doi.org/10.7910/DVN/D8HZLJ. To visualise the distribution of the raw protein intensities as histograms, box and PCA plots use this link https://mudiboevans.shinyapps.io/shinyapp/.

## Code availability
The analysis codes used in this study are available at Havard Dataverse repository[74] through this link https://doi.org/10.7910/DVN/D8HZLJ. Similar version of the analysis codes have been deposited at HIV-SM-PROTEOMICS GitHub repository[84].

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

## Acknowledgements

We thank the CHAIN Network study for their invaluable contribution to the current study, including provision of data. This work was supported by the Bill & Melinda Gates Foundation (INV-000791) and (OPP1131320) awarded to JAB and JLW. For the purpose of open access, the CHAIN Network has applied a CC BY public copyright license to any author-accepted manuscript version arising from this submission. The study was also supported by theNational Institute for Health and Care Research (NIHR201813) awarded to A.J.P., G.B.G., P.K., J.A.B., K.D.T., J.M.N., and B.O.S.; and Medical Research Council–Department for International Development–Wellcome Trust Joint Global Health Trials scheme (MR/M007367/1) awarded to J.A.B. E.O.M. is a PhD fellow at KEMRI-Wellcome Trust and Wageningen University funded by National Institute for Health Research (NIHR201813). A.J.P. is funded by Wellcome (108065/Z/15/Z). J.M.N. is supported by a Wellcome Trust Intermediate Fellowship (222967/B/21/Z). The funders had no role in the design, conduct, analysis or writing of this manuscript.

## Author contributions

J.M.N., K.D.T., and M.M.N. designed the nested case-cohort study within CHAIN Network cohort. E.O.M. designed case-control study within CHAIN, performed the statistical analysis and wrote the first manuscript draft. J.M.N. and G.B.G. designed the current study, provided

supervision and advice on design, analysis, interpretation of the results and critically reviewed the manuscript. These authors contributed equally: J.M.N. and G.B.G. J.A.B. provided advice on design, analysis, result interpretation and critically reviewed the manuscript. M.M.N., E.K., and N.N. managed data. C.T., M.T., J.T., R. Musyimi., E.O., E.Mbale., J.M., R.M.B., A.H.D., B.O.S., and E.Mupere coordinated sample collection and shipment. R.H.J.B., J.L.W., A.J.P., K.D.T., C.L.L., C.J.M., and P.K. critically reviewed the manuscript. J.B. provided advice and reviewed structural equation models. J.A.B. and J.L.W. principal investigators of the original CHAIN Network study. All authors approved the manuscript and had final responsibility for the decision to submit for publication.

## Competing interests

The authors declare no competing interests.

## Additional information

**Evans O. Mudibo** [1,2,3,4] ✉, **Jasper Bogaert** [5], **Caroline Tigoi**[1,3], **Moses M. Ngari** [1,3], **Benson O. Singa**[3,6], **Christina L. Lancioni** [3,7], **Abdoulaye Hama Diallo**[8,9], **Emmie Mbale**[10], **Ezekiel Mupere** [11], **John Mukisa**[12], **Johnstone Thitiri**[1,3], **Molline Timbwa**[1,3], **Elisha Omer**[1,3], **Narshion Ngao**[1,3], **Robert Musyimi**[1,3], **Eunice Kahindi**[1,3], **Roseline Maïmouna Bamouni**[8], **Robert H. J. Bandsma** [13,14], **Paul Kelly** [15,16], **Andrew J. Prendergast** [15,17], **Christine J. McGrath**[3,18], **Kirkby D. Tickell**[3,18], **Judd L. Walson** [3,19], **James A. Berkley** [1,3,20], **James M. Njunge** [1,3,21] ✉ & **Gerard Bryan Gonzales** [2,3,4,21] ✉

[1]KEMRI-Wellcome Trust Research Programme, Kilifi, Kenya. [2]Division of Human Nutrition and Health, Wageningen University & Research, Wageningen, The Netherlands. [3]The Childhood Acute Illness and Nutrition Network, Nairobi, Kenya. [4]Department of Public Health and Primary Care, Faculty of Medicine and Health Sciences, Ghent University, Ghent, Belgium. [5]Department of Data Analysis, Faculty of Psychology and Educational Sciences, Ghent University, Ghent, Belgium. [6]Center for Clinical Research, Kenya Medical Research Institute, Nairobi, Kenya. [7]Department of Pediatrics, Oregon Health and Science University, Portland, OR, USA. [8]Department of Public Health, University Joseph Ki-Zerbo, Ouagadougou, Burkina Faso. [9]Department of Public Health, Centre Muraz Research Institute, Bobo-Dioulasso, Burkina Faso. [10]Department of Paediatrics and Child Health, Kamuzu University of Health Sciences, Blantyre, Malawi. [11]Department of Paediatrics and Child Health, Makerere University College of Health Sciences, Kampala, Uganda. [12]Department of Immunology and Department of Molecular Biology Makerere University College of Health Sciences, Kampala, Uganda. [13]Translational Medicine Program, Research Institute, Hospital for Sick Children, Toronto, Canada. [14]Department of Nutritional Sciences, Faculty of Medicine, University of Toronto, Toronto, ON, Canada. [15]Blizard Institute, Queen Mary University of London, London, UK. [16]Department of Medicine, Tropical Gastroenterology and Nutrition Group, University of Zambia School of Medicine, Lusaka, Zambia. [17]Zvitambo Institute for Maternal and Child Health Research, Harare, Zimbabwe. [18]Department of Global Health, University of Washington, Seattle, WA, USA. [19]Departments of International Health, Pediatrics and Medicine, Bloomberg School of Public Health, Johns Hopkins University, Baltimore, MD, USA. [20]Nuffield Department of Medicine, University of Oxford, Oxford, UK. [21]These authors contributed equally: James M. Njunge, Gerard Bryan Gonzales. ✉e-mail: EMudibo@kemri-wellcome.org; JNjunge@kemri-wellcome.org; Bryan.Gonzales@UGent.be

