## [Peer Review file · Nature Communications]

Systemic biological mechanisms underpin poor post-discharge growth among severely wasted children with HIV

Corresponding Author: Mr Evans Mudibo

Version 0:

Reviewer comments:

Reviewer #1

(Remarks to the Author)

This is a case-control study conducted using data from the Childhood Acute Illness and Nutrition (CHAIN) Network cohort study that seeks to evaluate proteomic signatures associated with HIV and their impact on growth after hospital discharge in children who are admitted to the hospital for acute illness and also have severe wasting. Data are included from 4 countries in Sub-Saharan Africa (data from Asian countries excluded). This is an important study that identifies potential biological processes/pathways that may link HIV to early post-discharge growth among severely malnourished children in sub-Saharan Africa.

The study presents interesting results, but seems to have two different goals - studying the effect of HIV on post-discharge growth through 180 days, and then also the mechanistic questions surrounding proteomics for severe wasting. The two goals are not prioritized consistently throughout the manuscript. Would suggest prioritizing both goals, or just focusing on the latter. This is my major suggestion for revision.

Introduction

-Largely focuses on severe malnutrition; however subsequent analyses are not only focused on the group of children with severe wasting. Please make consistent.

Methods

-It is confusing which analyses are conducted on the broader group of children with acute illness vs. the group of children with severe wasting. Please improve clarity in the manuscript (analysis vs. mechanistic analysis is confusing, and not all analyses are mentioned in the title and abstract)

-What were reasons for why children were admitted to hospital?

-Line 177-178 - were these clinical syndromes at admission or discharge? please clarify the timing of these variables

Results

-It would be useful to see the characteristics of participants with severe wasting who were selected for the mechanistic analyses. Also, how did these participants differ from the ones with severe wasting in Table 1? They survived up to 3 months and had proteome data, but it would be useful to the reader to see the characteristics for the different groups being considered for different analyses.

Lines 289-291: were any formal statistical tests conducted to test the "implication" of worse post-discharge trajectories?

Discussion

-Do the authors have any comments on why findings are reported in relation to certain growth parameters but not others?

(Remarks on code availability)

Reviewer #2

(Remarks to the Author)

Mudibo et al. utilized plasma proteomics data measured by the SomaScan® assay to study the potential mechanisms about how HIV influences post-discharge growth in children with severe malnutrition (SM). Overall, the data analysis was comprehensive and well performed. Below are my comments regarding the current manuscript:

Major points:

- 1) The proteomics dataset used in this study needs clearer presentation. The manuscript describes the selection of cases and the generation of proteomics data separately, but it is unclear which data (e.g., 7335 proteins × ? cases) were used for the "Comparative HIV analysis" and which for the "HIV mechanistic analysis" (as shown in Figure 1A). Additionally, in large-scale studies with hundreds of samples, missing values can be a significant issue. Were all 7335 proteins quantified across all samples? If not, the authors should provide information on the percentage of missing values and any strategies employed to address this issue.
- 2) The proteomics dataset is not available for review, which is understandable due to potential restrictions. However, to give readers a better understanding of the data, some basic QC plots should be included. Examples include a histogram to show the overall intensity distribution, a boxplot for the intensity distribution of each sample, and PCA plots to indicate whether samples cluster based on biological groups or if covariates such as sex and age influence protein abundances.
- 3) Equation 2 appears incomplete; it seems to lack a coefficient before the HIV status variable. Is the X-axis of Figure 2A representing this coefficient? Additionally, the term "weighted" in equation 2 requires further clarification. The manuscript states, "Weight ... was generated by computing the probability of a child being HIV positive given their age, sex, nutritional status, clinical syndromes (malaria, diarrhea, pneumonia), and site of recruitment." However, how exactly this weight was calculated remains unclear. Providing an equation or further explanation for this process would be helpful.
- 4) The databases used for biological pathway analysis are somewhat outdated. For instance, DAVID v6.8 was released in 2016, while a more recent version is available with quarterly updates. WebGestalt also released a newer version this year. STRING v12 is up-to-date, which is fine. The authors should briefly explain their choice of these databases and clarify whether all three were used for the same purpose (e.g., GO biological process enrichment). If so, how were the results from different databases integrated? If the databases were used for different purposes, this should be clearly stated.
- 5) The section on "Hub proteins identification" in the methods needs more detail:
 - (A) How was the association between each protein in a module and HIV status calculated? Was it done using the same method as in equation 2?
 - (B) How was "module membership" used in subsequent analysis? Is it equivalent to connectivity Ki?
 - (C) What exactly is meant by "The absolute effect sizes of protein-HIV significance"? Is this the beta-coefficient from equation 2?
 - (D) How were hub proteins selected? Based on Figure 3, it seems that scaled intra-modular connectivity was the primary determinant for selecting hub proteins.
- 6) The SEM analysis results indicate that "Age at discharge" is significantly associated with modules in all phenotypes (MUAC, WAZ, WHZ, and HAZ). What is the biological explanation for this finding? Should this effect have been accounted for at the beginning of the data analysis to prevent potential confounding?

Minor points:

- 7) The quality of the figures could be further improved. For instance, the sizes of the scatter plots in Figure 4 vary, which could be standardized for consistency.

(Remarks on code availability)

Version 1:

Reviewer comments:

Reviewer #2

(Remarks to the Author)

The authors have addressed all my concerns.

(Remarks on code availability)

Reviewer #4

(Remarks to the Author)

(Remarks on code availability)

NA

Dear Editor and Reviewers,

We would like to sincerely thank you and the reviewers for your time, thoughtful comments, and valuable feedback on our manuscript, Systemic biological mechanisms underpin poor post-discharge growth among severely wasted children with HIV (Manuscript number: NCOMMS-24-42068-T). We greatly appreciate the constructive suggestions and insights, which have significantly improved the quality and clarity of our work.

We have carefully considered each comment and have made substantial revisions to address all points raised. Below is a detailed, point-by-point response to the reviewers' comments, along with corresponding changes made in the manuscript.

We hope that these revisions meet your expectations, and we look forward to your feedback.

Reviewer #1 (Remarks to the Author):

This is a case-control study conducted using data from the Childhood Acute Illness and Nutrition (CHAIN) Network cohort study that seeks to evaluate proteomic signatures associated with HIV and their impact on growth after hospital discharge in children who are admitted to the hospital for acute illness and also have severe wasting. Data are included from 4 countries in Sub-Saharan Africa (data from Asian countries excluded). This is an important study that identifies potential biological processes/pathways that may link HIV to early post-discharge growth among severely malnourished children in sub-Saharan Africa.

Major comment: *The study presents interesting results, but seems to have two different goals - studying the effect of HIV on post-discharge growth through 180 days, and then also the mechanistic questions surrounding proteomics for severe wasting. The two goals are not prioritized consistently throughout the manuscript. Would suggest prioritizing both goals, or just focusing on the latter. This is my major suggestion for revision.*

Response: We thank the reviewer for this important feedback. We have revised the manuscript to highlight the main priorities of our study. Our manuscript focuses on three main complimentary points: (1) the effect of HIV on six months post-discharge growth among children with severe malnutrition and those at risk of malnutrition, (2) the systemic pathways associated with HIV in these children, and (3) how these HIV-associated biological processes impacts post-discharge growth among children with severe malnutrition. Our approach follows a sequential strategy, beginning with a broader group of children to identify relevant pathways, which then inform our deeper analysis of growth-related pathways in children with HIV-SM.

Changes made: The manuscript has been revised as follows; abstract on lines 52-55 and 58-59, introduction on line 111-115, results on lines 165-169 and 283-285, discussion on line 319-322 and line 330 on page 8, and methods line 500-502.

Other comments

Introduction

Comment 1: *Largely focuses on severe malnutrition; however subsequent analyses are not only focused on the group of children with severe wasting. Please make consistent.*

Response: Thank you for highlighting this point. As explained earlier, we employed a sequential strategy to enable targeted analysis of specific pathways linked to post-hospitalisation growth in children with HIV-SM. The reviewer is correct in noting that our study included both children with SM and those at risk of SM.

Our initial step was to examine the effects of HIV on growth six months post-hospitalisation. Once we confirmed HIV's impact, we focused on identifying HIV-associated biological processes in children with severe malnutrition or at risk of malnutrition following recovery from acute illness. This allowed us to prioritize pathways for deeper analysis, where we applied structural equation modeling to explore how HIV-related biological processes influence post-hospitalisation growth. By this approach, we limited our analysis to proteins specifically related to HIV, allowing us to construct a causal model more effectively.

Nonetheless, we have now revised both the introduction and discussion to provide a more balanced presentation of malnutrition, explicitly addressing wasting, underweight, and stunting alongside severe wasting, ensuring consistency throughout the manuscript.

Changes made: The introduction has been revised on lines 76-86 and discussion section line 344-349.

Methods

Comment 2: *It is confusing which analyses are conducted on the broader group of children with acute illness vs. the group of children with severe wasting. Please improve clarity in the manuscript (analysis vs. mechanistic analysis is confusing, and not all analyses are mentioned in the title and abstract).*

Response: We regret that this has caused confusion. Hence, we have revised the methods section to explicitly clarify the number of children included in each analysis. We now provide a clear breakdown of how many children were included in the comparative and mechanistic analyses. Additionally, we have clarified the sets of analyses to show that we first determined effect of HIV on six months post-discharge growth, identified the systemic processes associated with HIV and lastly investigated how these HIV-associated systemic processes influences growth among severely malnourished children. We believe that our abstract and title convey key methods used in the study. We also believe that our title is precise and concise enough to attract the attention of the readers.

Changes made: The methods section on page 11, line 507-509 and line 514 on page 12, and Figure 1A on page 23, have been revised to clarify the comparative and mechanistic analyses, and the number of children included in each analysis. Further in Table 1 and Table 3 on pages 31 and 33 respectively, and in Supplementary Table 1 we have mentioned the number of children included in each analysis.

Comment 3: *What were reasons for why children were admitted to hospital?*

Response: Thank you for highlighting this. The methods section has been revised to include clinical syndromes for hospital admission.

Changes made: Methods section on page 11, line 486-488 highlights some of the clinical symptoms children presented with.

Comment 4: *Line 177-178 - were these clinical syndromes at admission or discharge? please clarify the timing of these variables.*

Response: Thank you for highlighting this. The methods section has been revised to explicitly specify that the clinical syndromes were noted at admission to hospital.

Changes made: Methods section on page 13, line 591 specifies that the clinical syndromes were at admission.

Results

Comment 5: *It would be useful to see the characteristics of participants with severe wasting who were selected for the mechanistic analyses. Also, how did these participants differ from the ones with severe wasting in Table 1? They survived up to 3 months and had proteome data, but it would be useful to the reader to see the characteristics for the different groups being considered for different analyses.*

Response: We thank the reviewer for pointing this out. Baseline characteristics of the participants included in the mechanistic analyses were initially in the Supplementary Information, which we now included in the main manuscript file. These characteristics are detailed in Table 3. Children included in the mechanistic analysis (Table 3) comes out of a case-cohort design which was designed to be representative of the initial population. In the current analysis, our controls (children without HIV) are from a random selection of the cohort (line 503-504), an indication that the current study was designed to be representative.

Changes made: Results section on page 6, line 241 cites Table 3 that details baseline characteristics of children included in the mechanistic analysis. Table 3 is found on page 33.

Comment 6: *Lines 289-291: were any formal statistical tests conducted to test the "implication" of worse post-discharge trajectories?*

Response: We regret that the use of the word "implying" has caused confusion. The statement was meant to mean that "the study children had mean monthly losses of 0.06 (95% CI: -0.08 to -0.04) HAZ during the follow up period". We have now clarified this in the revised manuscript.

Changes made: Results section on page 4, line 160 specifies that the study children had mean monthly losses of 0.06 (95% CI: -0.08 to -0.04) HAZ during the follow up period.

Discussion

Comment 7: *Do the authors have any comments on why findings are reported in relation to certain growth parameters but not others?*

Response: We appreciate this observation. Our study focused on MUAC, WAZ and WHZ as these are the World Health Organisation (WHO) criteria for diagnosing malnutrition in children under 5 years old¹. Other growth parameters not directly linked to this diagnostic criteria were not included. However, to enrich the manuscript we have provided a two-paragraph discussion on the HAZ findings.

Changes made: We have revised the discussion section paragraph 1, line 322 on page 7 and line 330 on page 8 to specify that our primary focus was MUAC, WAZ and WHZ based on the WHO criteria of diagnosing malnutrition in children under 5 years. Additionally, we have added two paragraphs to discuss findings attributed with HAZ, lines 435-451.

Reviewer #2 (Remarks to the Author):

Mudibo et al. utilized plasma proteomics data measured by the SomaScan™ assay to study the potential mechanisms about how HIV influences post-discharge growth in children with severe

malnutrition (SM). Overall, the data analysis was comprehensive and well performed. Below are my comments regarding the current manuscript:

Major points:

Comment 1): *The proteomics dataset used in this study needs clearer presentation. The manuscript describes the selection of cases and the generation of proteomics data separately, but it is unclear which data (e.g., 7335 proteins × ? cases) were used for the “Comparative HIV analysis” and which for the “HIV mechanistic analysis” (as shown in Figure 1A). Additionally, in large-scale studies with hundreds of samples, missing values can be a significant issue. Were all 7335 proteins quantified across all samples? If not, the authors should provide information on the percentage of missing values and any strategies employed to address this issue.*

Response: We appreciate this observation. In response, we have revised the methods section to explicitly clarify the number of children included in each analysis. We now provide a clear breakdown of how many children were included in the comparative and mechanistic analyses. Additionally, we have clarified the sets of analyses conducted in the study. Plasma proteins were quantified in all 689 children that were selected. Supplementary Table 1 highlights the characteristics of these children. Across all these children, data from the SomaScan™ assay which contained all 7335 proteins quantified was provided by SomaLogic Inc, and did not contain missing data. The data underwent normalization and standardization algorithm corrections using the SomaLogic proprietary pipeline. Other studies that utilised SomaScan™ technology from SomaLogic Inc also noted no missing values, and it has been noted that the failure to report missing values is one of the remarkable characteristics of SomaScan™ sensitivity².

Changes made: Methods section on page 11, line 507-509, page 12 line 514-516 and Figure 1A on page 23 explicitly clarifies comparative and mechanistic analyses, and the number of children included in each analysis. Further Tables 1, 3 and Supplementary Table 1 summarise the numbers of children included in each analysis alongside their baseline characteristics.

Comment 2): *The proteomics dataset is not available for review, which is understandable due to potential restrictions. However, to give readers a better understanding of the data, some basic QC plots should be included. Examples include a histogram to show the overall intensity distribution, a boxplot for the intensity distribution of each sample, and PCA plots to indicate whether samples cluster based on biological groups or if covariates such as sex and age influence protein abundances.*

Response: Thank you for this observation. The data used in this study is already published with a citable DOI (<https://doi.org/10.7910/DVN/D8HZLJ>), manuscript line 697. To access the data a formal request should be sent to the data governance committee. Instructions on how to access this data is clearly stated in the manuscript under data availability section, line 696-700. However, based on the reviewer’s important comment, we have now included a web link for visualizing raw protein intensities, which allows readers to explore our raw data and built box plots, histograms and PCA plots.

Changes made: The methods section on page 12, line 534-535 clarifies that the data provided from the SomaScan™ assay did not contain missing values thus, no data imputation was done. Also, in the main manuscript file we have provided this link <https://mudiboevans.shinyapps.io/shinyapp/> for visualisation of the raw protein intensities, line 537 on page 12 and line 701 on page 16. Through this link you can visualise the distribution of the raw protein intensities as boxplots, histograms and PCA

plots. For PCA and box plots the distribution can be visualised by sex (biological attribute), site of enrolment, nutritional (no wasting, moderate wasting and severe wasting) or HIV status.

Comment 3): *Equation 2 appears incomplete; it seems to lack a coefficient before the HIV status variable. Is the X-axis of Figure 2A representing this coefficient? Additionally, the term "weighted" in equation 2 requires further clarification. The manuscript states, "Weight ... was generated by computing the probability of a child being HIV positive given their age, sex, nutritional status, clinical syndromes (malaria, diarrhea, pneumonia), and site of recruitment." However, how exactly this weight was calculated remains unclear. Providing an equation or further explanation for this process would be helpful.*

Response: Thank you for this important comment. We have revised equation 2 to include the beta coefficient before HIV status and we have added further details explaining how weights were generated using inverse probability weighting (IPW). IPW is a well-established and validated method in causal inference and epidemiology^{3,4}, which is used to adjust for confounders especially when the confounding structure is more complex (i.e., certain design considerations like in nested case-control studies like our study). The Supplementary Methods has now been updated with Supplementary equations 1.0-1.2 included to explain the generation of the weights. In the codes we also provided the R script used to generate the weights using inverse probability weighting. Regarding the question about Figure 2A, the X-axis indeed represents the beta-coefficients of the association between HIV status and protein modules.

Changes made: Methods section on page 14, line 599 we have added a coefficient before HIV status. On line 592-594 we have added a link (<https://doi.org/10.7910/DVN/D8HZLJ>) and Supplementary Methods where codes and equations for computing IPW can be found. On line 603 we have clarified that the term "weighted" as used in equation 2 refers to the inclusion of weights generated using IPW.

Comment 4): *The databases used for biological pathway analysis are somewhat outdated. For instance, DAVID v6.8 was released in 2016, while a more recent version is available with quarterly updates. WebGestalt also released a newer version this year. STRING v12 is up-to-date, which is fine. The authors should briefly explain their choice of these databases and clarify whether all three were used for the same purpose (e.g., GO biological process enrichment). If so, how were the results from different databases integrated? If the databases were used for different purposes, this should be clearly stated.*

Response: We thank you for this observation. We used the online version of these databases (DAVID, WebGestalt and STRING) for GO biological process enrichment analysis. The version of DAVID (v6.8) quoted in the initial version of manuscript was misleading. We have corrected the version of the DAVID database we used to DAVID v2023q4. For WebGestalt database, at the time of our analysis the available version was the 2019 version. The current WebGestalt 2024 version was published on May 29, 2024⁵. We have clarified that we utilized the three databases independently to explore biological processes. Different algorithms are implemented by different databases to compute fold enrichment and infer enriched GO terms, thus utilising several databases provides more information. The results from each of the three databases we used were consistent, providing comprehensive insights (Table 2, Supplementary Table 2, 3 and 5).

Changes made: Methods section on page 14, line 608-611 now clarifies that we used online databases and provides updated DAVID v2023q4 release that we used. Line 615-616 clarifies that the three databases were used independently thus, no integration was done.

5): The section on “Hub proteins identification” in the methods needs more detail

Comment (A): *How was the association between each protein in a module and HIV status calculated? Was it done using the same method as in equation 2?*

Response: We thank you for highlighting this. The association between proteins in a module and HIV status was not computed using equation 2. We have clarified that we calculated this association by correlating expression profile of each protein in a module with HIV status. From this correlation we obtained protein-HIV significance whose absolute values and intra-modular connectivity were used in the determination of hub proteins. This is a well documented method in WGCNA⁶ R package. We implemented this method as specified by the authors of the package.

Changes made: Page 14, line 625-627, clarifies how the association between each protein in a module and HIV status was calculated.

Comment (B): *How was “module membership” used in subsequent analysis? Is it equivalent to connectivity Ki?*

Response: Thank you for noting this out. Module membership (kME) is the correlation between the expression profile of a protein and the module eigenprotein (the first principal component of the module). It measures how closely a protein's expression profile matches the overall expression pattern of the module. Besides intra-modular connectivity (Ki), kME can also be used to identify hub proteins. Module membership is not equivalent to Ki (Ki refers to the sum of connection strengths (adjacencies) of a protein with all other proteins within the same module). In this analysis, for hub protein identification we only used intra-modular connectivity and not module membership. The unclear “module membership” statement has been removed in the revised manuscript.

Changes made: Page 14 line 628-631, clarifies that intra-modular connectivity was used instead of module membership.

Comment (C): *What exactly is meant by “The absolute effect sizes of protein-HIV significance”? Is this the beta-coefficient from equation 2?*

Response: We appreciate the reviewer for this observation. “The absolute effect sizes of protein-HIV significance” is the absolute value (magnitude) of the correlation between expression profile of every protein in a module and HIV status and it is not the beta-coefficient from equation 2. The beta-coefficient from equation 2 is an estimate of association between the first principal component of the module with HIV status. Computation of significance between biological molecules e.g., genes/proteins and disease trait is an established method in WGCNA⁶ R package.

Changes made: Page 14 line 625-627, clarifies what “The absolute effect sizes of protein-HIV significance” is and how it was computed.

Comment (D): *How were hub proteins selected? Based on Figure 3, it seems that scaled intra-modular connectivity was the primary determinant for selecting hub proteins.*

Response: We appreciate the reviewer for this comment. Hub proteins were identified by determining the association between absolute effect size of protein-HIV significance and intra-modular connectivity

which defines the strength of adjacencies of a protein with all other proteins in a module. The slope of the regression line of this association gives the hub protein significance. In a module, a protein(s) with high intra-module connectivity is well-integrated into the module and interacts strongly with other proteins in that module. Both absolute effect sizes of protein-HIV significance and intra-modular connectivity are important in determining potential hub proteins driving network associated with HIV for example in Figure 3. In a module there could be more than one hub protein, we have provided an elaborate list of top candidate hub proteins with high effect sizes and intra-modular connectivity in Supplementary Table 4 for modules associated with HIV status.

Changes made: Methods section page 14, line 625-634, specifies hub protein identification.

Comment 6): *The SEM analysis results indicate that “Age at discharge” is significantly associated with modules in all phenotypes (MUAC, WAZ, WHZ, and HAZ). What is the biological explanation for this finding? Should this effect have been accounted for at the beginning of the data analysis to prevent potential confounding?*

Response: Thank you for pointing this out. There is indeed a significant association of age with the modules associated with MUAC, WAZ, WHZ and HAZ suggesting that age is indeed a confounder. To account for this effect in our analyses we always adjusted for age as indicated in our models in the manuscript. Specifically the following strategies were used:

- a) We included age as one of the covariates in the generation of weights using inverse probability weighting (IPW) to balance the baseline characteristics used in equation 2 as stated in method section, line 586-592.
- b) We adjusted for age in equations 3, line 659-660.
- c) We adjusted for age in the linear models included in the SEM analysis as stated in line 675-677 of the manuscript file. The R codes used in SEM analysis are found through the link available in the code availability section, line 704.

Minor points:

Comment 7): *The quality of the figures could be further improved. For instance, the sizes of the scatter plots in Figure 4 vary, which could be standardized for consistency.*

Response: We appreciate this observation. Clearer images with consistent sizes have been generated, Figure 1, 2, 4 and 5 have been revised.

Reference:

- 1 WHO. WHO guideline on the prevention and management of wasting and nutritional oedema (acute malnutrition) in infants and children under 5 years. (2023).
- 2 Candia, J., Daya, G. N., Tanaka, T., Ferrucci, L. & Walker, K. A. Assessment of variability in the plasma 7k SomaScan proteomics assay. *Sci Rep* 12, 17147 (2022). <https://doi.org:10.1038/s41598-022-22116-0>
- 3 Chesnaye, N. C. et al. An introduction to inverse probability of treatment weighting in observational research. *Clin Kidney J* 15, 14-20 (2022). <https://doi.org:10.1093/ckj/sfab158>
- 4 SAMUELSEN, S. O. A pseudolikelihood approach to analysis of nested case-control studies. *Biometrika* 84, 379-394 (1997). <https://doi.org:10.1093/biomet/84.2.379>
- 5 Elizarraras, J. M. et al. WebGestalt 2024: faster gene set analysis and new support for metabolomics and multi-omics. *Nucleic Acids Research* 52, W415-W421 (2024). <https://doi.org:10.1093/nar/gkae456>
- 6 Langfelder, P. & Horvath, S. WGCNA: an R package for weighted correlation network analysis. *BMC Bioinformatics* 9, 559 (2008). <https://doi.org:10.1186/1471-2105-9-559>